# Endowing homodimeric carbamoyltransferase GdmN with iterative functions through structural characterization and mechanistic studies

Jianhua Wei [1,2,4], Xuan Zhang[3,4], Yucong Zhou[1,2], Xingnuo Cheng[1,2], Zhi Lin[1,2], Mancheng Tang [1,2], Jianting Zheng [1,2], Binju Wang [3], Qianjin Kang [1,2] ✉ & Linquan Bai [1,2] ✉

Iterative enzymes, which catalyze sequential reactions, have the potential to improve the atom economy and diversity of industrial enzymatic processes. Redesigning one-step enzymes to be iterative biocatalysts could further enhance these processes. Carbamoyltransferases (CTases) catalyze carbamoylation, an important modification for the bioactivity of many secondary metabolites with pharmaceutical applications. To generate an iterative CTase, we determine the X-ray structure of GdmN, a one-step CTase involved in ansamycin biosynthesis. GdmN forms a face-to-face homodimer through unusual C-terminal domains, a previously unknown functional form for CTases. Structural determination of GdmN complexed with multiple intermediates elucidates the carbamoylation process and identifies key binding residues within a spacious substrate-binding pocket. Further structural and computational analyses enable multi-site enzyme engineering, resulting in an iterative CTase with the capacity for successive 7-*O* and 3-*O* carbamoylations. Our findings reveal a subclade of the CTase family and exemplify the potential of protein engineering for generating iterative enzymes.

Iterative enzymes catalyze two or more modification steps in natural product biosynthesis and have intriguing potential for the development of novel drugs and other chemicals with structural complexity and new biological activities. Known iterative enzymes include cytochrome P450 enzymes[1–3], methyltransferases[4], glycosyltransferases[5–7], vanadium-dependent haloperoxidase[8], and flavin-dependent halogenase[9]. Usually, enzymes offer distinctive advantages over chemical catalysts due to their chemo-, stereo- and/or regioselectivities, and enzyme-catalyzed reactions can be carried out under routine laboratory conditions, with high-performance productivity and convenient downstream purification of the target molecules. Moreover, compared with a series of canonical enzymes that can each only catalyze one step, iterative enzymes can provide catalytic versatility in synthetic biology, such as in the modification of therapeutic products. Therefore, repurposing of a canonical enzyme into an iterative biocatalyst is an attractive strategy for targeted generation of desired chemicals.

Carbamoylation, catalyzed by CTases, involves a direct or ATP-dependent transfer of a carbamoyl group from carbamoyl phosphate

[1]State Key Laboratory of Microbial Metabolism, Shanghai-Islamabad-Belgrade Joint Innovation Center on Antibacterial Resistances, School of Life Sciences and Biotechnology, Shanghai Jiao Tong University, Shanghai 200240, China. [2]Joint International Research Laboratory of Metabolic and Developmental Sciences, Shanghai Jiao Tong University, Shanghai 200240, China. [3]State Key Laboratory of Physical Chemistry of Solid Surfaces and Fujian Provincial Key Laboratory of Theoretical and Computational Chemistry, College of Chemistry and Chemical Engineering, Xiamen University, Xiamen 361005, China. [4]These authors contributed equally: Jianhua Wei, Xuan Zhang. ✉e-mail: qjkang@sjtu.edu.cn; bailq@sjtu.edu.cn

(CP) to an amino group, hydroxyl group, or thiol group of the recipient substrates (Fig. 1)[10–12]. CTases play fundamentally important roles in cellular metabolism and also participate in the biosynthesis of many bioactive secondary metabolites with pharmaceutical potentials[13–16]. Carbamoylated bioactive secondary metabolites are often found in the biosynthetic assemblies containing nucleosides, polyketide synthases (PKSs), PKS-non-ribosomal-peptide synthetase hybrids (PKS-NRPSs), and aminocoumarins (Supplementary Fig. 1).

CTases are also important tailoring enzymes for generating chemical diversity and improving the pharmacological properties of targeted compounds. For example, carbamoylated albicidin showed a significantly higher inhibitory activity against bacterial gyrase than albicidin did (-8 vs 49 nM)[17], and removal of the C7,9-cyclic carbinolamide group resulting from carbamoylation nearly abolished the antitumor biological activity of maytansinoids[18]. Maytansinoids are potent microtubule inhibitors and have served as the warhead molecules for antibody-drug conjugates for treating metastatic breast cancer[19]. Synthetic strategies for the antibody-drug conjugates developed so far include the initial preparation of maytansinol by chemical removal of the C-3 acyl group of ansamitocin under strict reaction conditions, which usually leads to low efficiency. Recently, 3-O-carbamoylation was reported to occur in ansacarbamitocins, with attachment by the CTase Asc21b from the *asc* cluster[20]. Dual carbamoylations at C-7 and C-3 of ansamitocins by engineered iterative CTases, instead

of with a multi-step chemical approach, would offer a more efficient, economical, and eco-friendly process for obtaining warhead drugs. Although structure-guided protein engineering can broaden substrate capacity[21], structural information on CTases, in terms of catalytic mechanisms and protein-substrate interactions, is limited.

Three main classes of CTases have been identified based on their catalytic mechanisms (Supplementary Fig. 2). Class I CTases, represented by aspartate transcarbamylase (ATCase), can transfer a carbamoyl group from CP directly to the amino group of an array of amino acid-derived substrates. The [NiFe]-hydrogenase maturation factor HypF represents the class II and engages in three half-reactions: HypF first generates carbamate derived from CP, followed by carbamoyladenylate (carbamoyl-AMP) formation in the presence of ATP and Mg$^{2+}$, and then it transfers the carbamoyl group from carbamoyl-AMP to the thiol of the C-terminal cysteine of HypE[15]. CTases in class III, which modify secondary metabolites, adopt two nucleophilic half-reactions, the formation of carbamoyl-AMP derived from ATP, CP, and Mg$^{2+}$ and the subsequent transfer of the carbamoyl group from carbamoyl-AMP to the secondary metabolites[16]. Further crystallographic investigations revealed the protein architectural differences of the different CTase classes (Fig. 1b and Supplementary Fig. 3). Class I CTases have a two-domain architecture, forming a general integrated fold with a homotrimer as the basic biological unit[22]. Class II CTases have four domains and function as monomers[15,23]. For class III CTases, only the

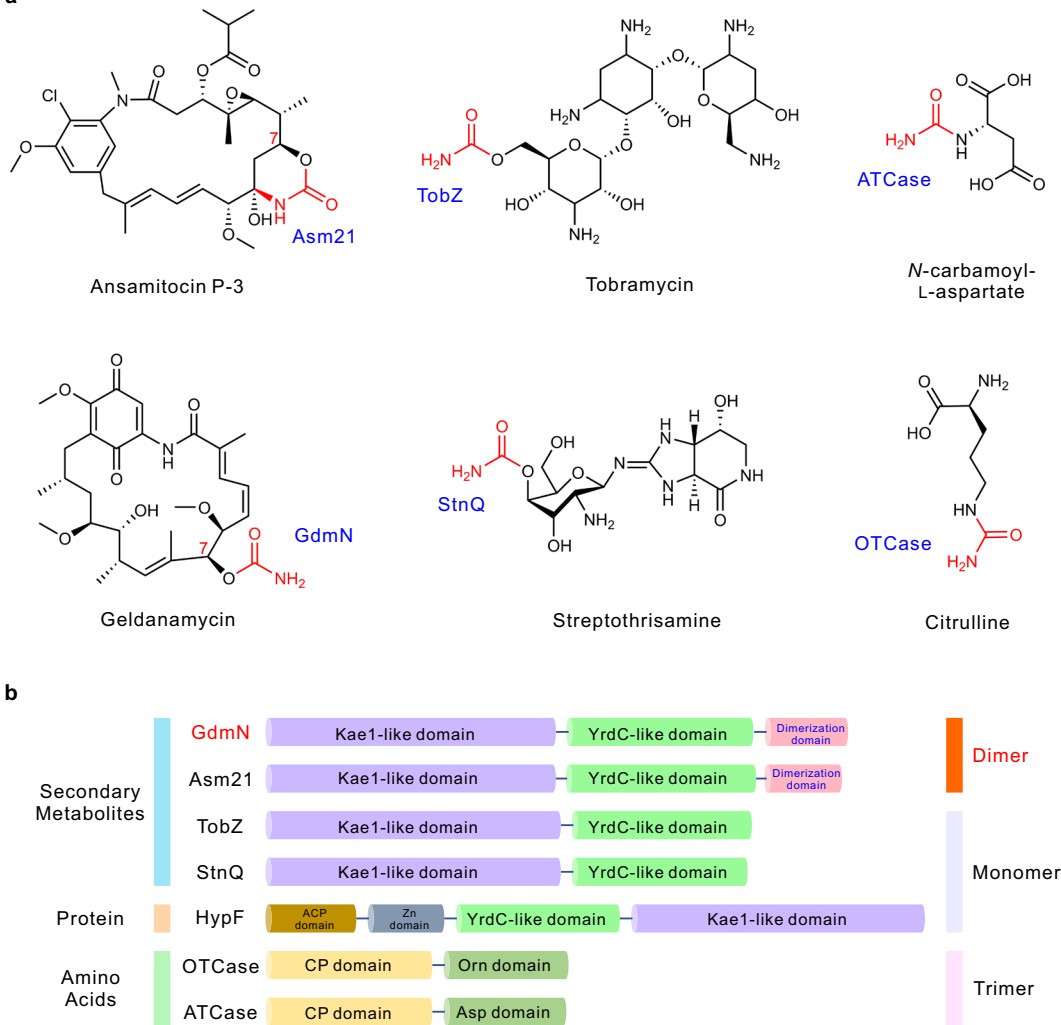

**Fig. 1 | Representative products and domain architecture of CTases. a** The various natural products catalyzed by CTases. The names of CTases are colored in blue and are located next to the carbamoyl group. **b** Substrate diversity, domain architecture, and aggregation states of representative CTases.

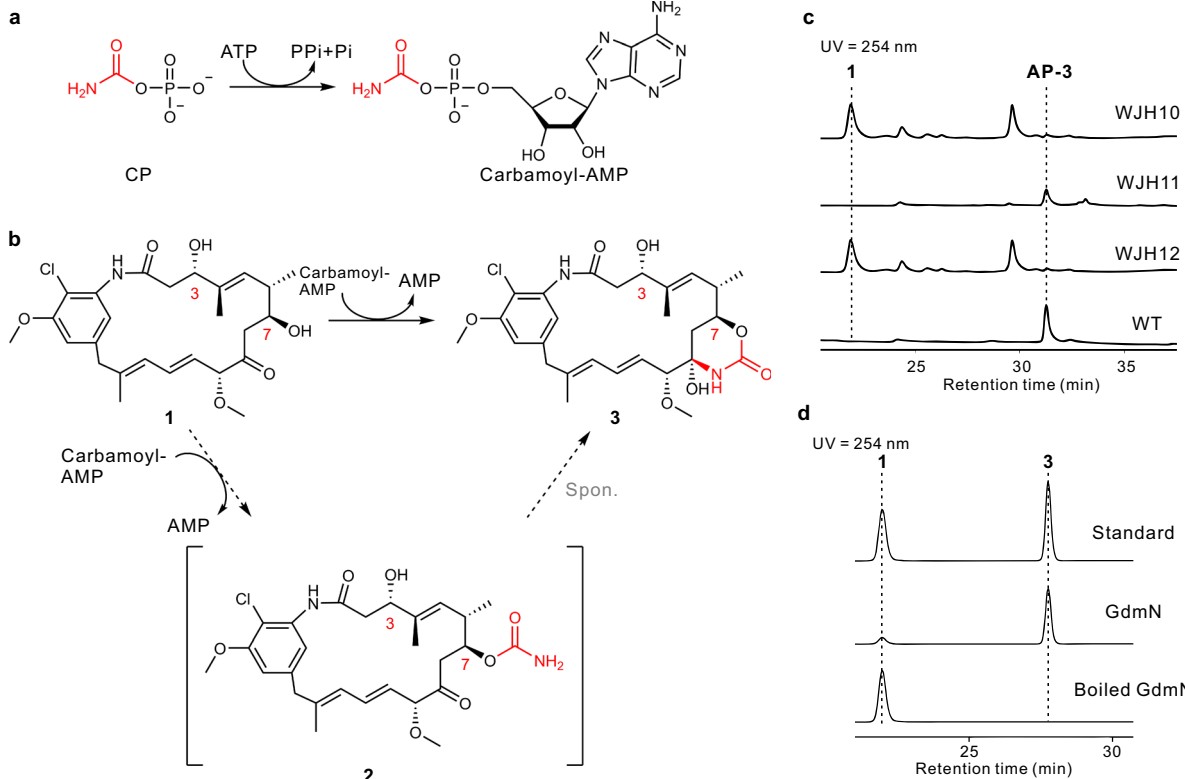

**Fig. 2 | Characterization of proposed known and predicted GdmN reactions.** **a** Scheme for the carbamoyl-AMP formation reaction occurring at the YrdC-like domain of GdmN. **b** Scheme for the carbamoyl group transfer reaction at the Kae1-like domain of GdmN. The solid line shows that the carbamoyl group transfers from carbamoyl-AMP to **1**, thereby generating **3** and AMP, and the dashed lines reveal that the proposed reaction in the solid line may consist of two half reactions, the enzymatic formation of **2** derived from carbamoyl group transfer and the spontaneous formation of **3** from **2**. **c** HPLC analysis of in vivo experiments of *gdmN* and truncated *gdmN*. WJH10, the CTase gene-disrupted mutant derived from *A. pretiosum* ATCC 31280; WJH11, WJH10 with *gdmN*; WJH12, WJH10 with truncated *gdmN*; WT, *A. pretiosum* ATCC 31280. **d** HPLC analysis of in vitro experiments catalyzed by GdmN in the presence of ATP, CP, and Mg$^{2+}$. Boiled GdmN is used as control.

crystallographic structure of TobZ, involved in aminoglycoside biosynthesis, has been determined, which indicated that TobZ possesses two catalytic domains and functions as a monomer[16]. Structures for the homodimeric CTases have not yet been reported.

In this study, we examine the molecular structure of the ansamycin-modifying CTase GdmN, with the goals of repurposing this enzyme for the iterative carbamoylation of ansamitocins. Our findings provide important advancements in the understanding of CTase function and also constitute an intriguing example of repurposing a canonical CTase into an iterative enzyme.

## Results

### Phylogenetic analysis revealed that ansamycin-modifying CTases belong to a subclade

Consistent with the division of their biological functions into three classes, a detailed phylogenetic analysis of CTases revealed three main clades (Supplementary Fig. 3). Class III CTases have a diverse membership involved in the assembly of products with a variety of chemical architectures (Supplementary Fig. 1). The holoenzymes of all characterized (class III) CTases are highly evolutionarily conserved with respect to their two nucleophilic conserved domains (Supplementary Fig. 4), comprising a catalytic domain (YrdC-like domain) for adenylation of CP with ATP and a catalytic domain (Kae1-like domain) for connection of the carbamoyl group with the recipients. Intriguingly, a sub-branch in class III was relatively independent from other members of this clade and consisted of the ansamycin-modifying CTases, including GdmN, Asm21, Asc21a, and Asc21b; as well as Orf7*[24], involved in concanamycin tailoring; and NovN[11,25,26], involved in novobiocin tailoring. Subsequent amino acid sequence alignment

determined that all members of this subclade contained an unusual auxiliary domain at the C-terminus, consisting of approximately 100 amino acids, which was absent from other classes of CTases (Fig. 1b and Supplementary Fig. 4). A Basic Local Alignment Search Tool (BLAST) search of these unique amino acids revealed no homology with known structures. The lack of structural information for this domain restricted our understanding of the quaternary structure of these CTases.

Our previous biochemical characterization had shown that Asm21 was capable of dual carbamoylations, utilizing both a polyketide backbone and a glycosyl moiety as substrates, during ansamitocin biosynthesis[27], suggesting that this subclade of CTases might contain a spacious binding pocket and hold promising potential for redesign. However, the absence of crystallographic structures for this subclade and their limited catalytic characterization impeded further rational design of these CTases for the iterative carbamoylations of ansamitocins.

### GdmN displayed substrate compatibility with Asm21 and presented an unusual homodimer form within the CTase family

GdmN and Asm21 contribute to *O*-carbamoylation at the C7 (*S* configuration) of geldanamycin and ansamitocins, respectively (Fig. 1a). These two enzymes share 60% sequence identity, indicating that GdmN may have a similar function to Asm21 in ansamitocin carbamoylation (Fig. 2a, b). To investigate the substrate compatibility of these two enzymes, constitutively overexpressed *gdmN* was inserted into the CTase gene-disrupted mutant WJH10, derived from the ansamitocin producer *Actinosynnema pretiosum* subsp. *pretiosum* ATCC 31280[28]. The resulting recombinant strain, WJH11, regained carbamoylation ability and

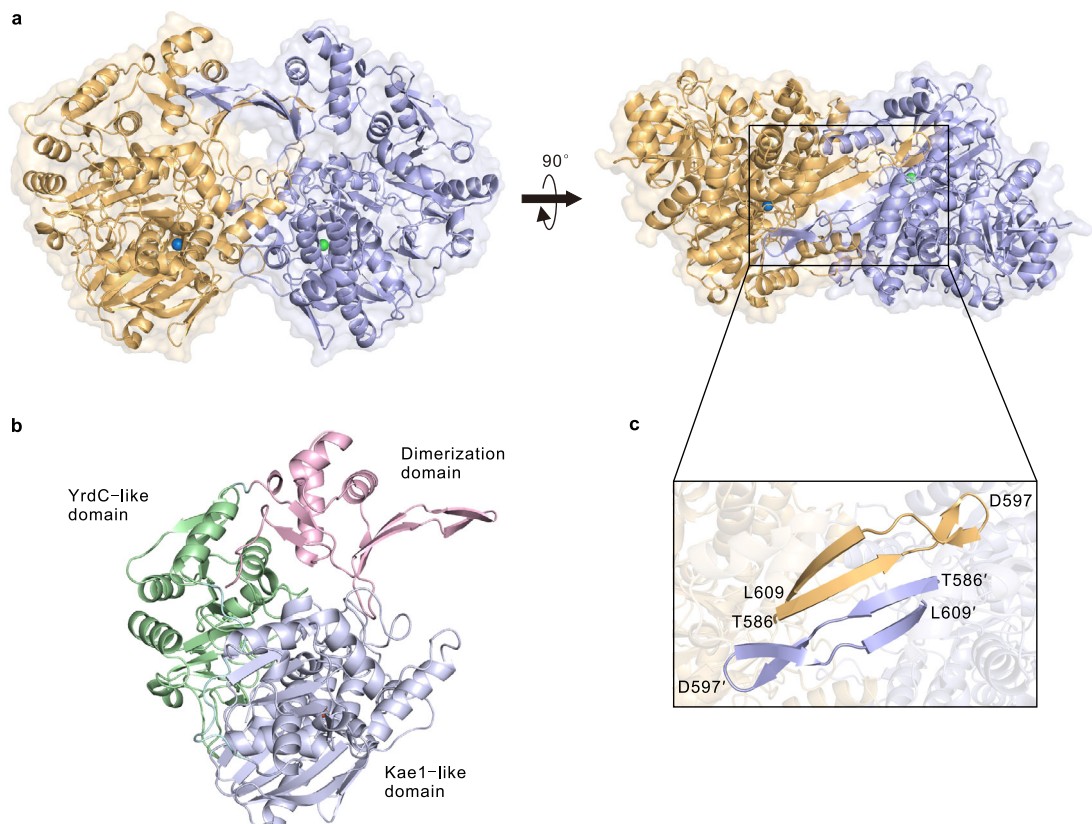

**Fig. 3 | Crystal structures of GdmN. a** The overall structure of GdmN. GdmN forms a homodimer architecture in the asymmetric unit, with different monomers colored in orange and violet. The iron ions are shown as blue and green spheres, respectively. Two bow-shaped monomers complement each other in a face-to-face fashion. **b** A GdmN monomer, with the Kae1-like domain, YrdC-like domain, and dimerization domain colored in violet, green, and pink, respectively. Cyan, linkers between the domains. **c** The antiparallel β-sheets in the dimerization domain are important for the formation of the homodimer.

recovered the production of ansamitocin P-3 (AP-3) (Fig. 2c and Supplementary Fig. 5). To determine the exact biochemical characteristics of GdmN, we obtained purified GdmN from *Escherichia coli* BL21(DE3) pLysS (Supplementary Fig. 6a). Using the preferred recipient substrate 20-*O*-methyl-19-chloroproansamitocin (**1**) of Asm21, we reconstituted the carbamoylation reaction in vitro for GdmN. Similar to the reactions of Asm21, the incubation of GdmN with 20-*O*-methyl-19-chloroproansamitocin (**1**), CP, $Mg^{2+}$, and ATP resulted in the specific appearance of *N*-demethyl-4,5-desepoxy-maytansinol (**3**) (Fig. 2d)[27]. These experiments demonstrated that GdmN can participate in ansamitocin assembly in the presence of ATP, CP, and $Mg^{2+}$ (Fig. 2a, b). Nevertheless, the complete catalytic mechanism, especially whether the formation of the C7,9-cyclic carbinolamide group is catalyzed by CTases, is still unknown (Fig. 2b).

Remarkably, size-exclusion chromatography analysis of GdmN determined that it has a homodimeric catalytic form in solution (Supplementary Fig. 7a), rather than the homotrimer defined for class I CTases or the monomer for TobZ and class II CTases. To investigate the role of the auxiliary domain anchoring the GdmN C-terminus, functional analysis was performed using truncated GdmN with a deletion in the C-terminal domain. The recombinant mutant WJH12, which constitutively overexpresses the truncated *gdmN* gene (M1-L575) in the CTase gene-disrupted mutant WJH10, did not accumulate AP-3 (Fig. 2c). Unfortunately, the GdmN mutant (M1-L575) was insoluble in vitro (Supplementary Fig. 6b) and so could not be tested. Nevertheless, the in vivo evidence suggested a significant contribution of the unique C-terminal domain for maintaining GdmN CTase architecture.

## X-ray structure revealed the C-terminal auxiliary domain is involved in the homodimer architecture formation of GdmN

To gain additional molecular insights, the X-ray crystal structure of *apo*-GdmN was determined with a resolution of 2.25 Å. Undoubtedly, consistent with the size-exclusion chromatography results, GdmN formed a homodimer architecture in the asymmetric unit (Fig. 3a and Supplementary Fig. 8). The monomer is similar to the other one, with an r.m.s.d. value of ~0.16 Å (Supplementary Fig. 8b). Each monomer formed a bow-shaped configuration, consisting of the auxiliary domain (D574-L674), namely the dimerization domain, located at the C-terminus; the canonical Kae1-like domain (M1-L351) at the N-terminus; and the YrdC-like domain (G368-A571) in the middle (Fig. 3b). The two bow-shaped monomers complemented each other in a face-to-face fashion, intertwining predominantly through the dimerization domain in addition to several interactions from the Kae1-like domain (Supplementary Table 1 and Supplementary Fig. 9), with a burying ~2,495 $Å^2$ surface area per monomer calculated by the PDBePISA server[29]. The C-terminal dimerization domain was composed of two central, two-stranded antiparallel β-sheets flanking three α-helices and three β-sheets. The central two-stranded antiparallel β-sheet (T586-L609) in each monomer built a four-stranded antiparallel β-sheet through the main chain hydrogen bonds, forming a vital geometry for the dimerization of GdmN (Fig. 3c and Supplementary Fig. 9c). The deletion of the β-sheets (the deletion of T586-D597 or T586-L609) in GdmN impeded the formation of the homodimer and abolished the activity, revealing that the GdmN quaternary structure maintained by β-sheets is necessary for GdmN function (Supplementary Fig. 7). Analysis of the dimerization domain in the DALI server[30] did not identify this fold in

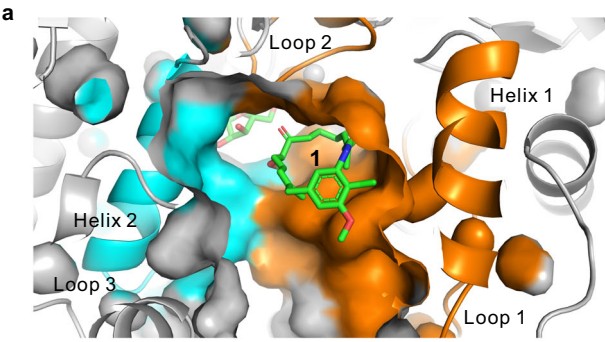

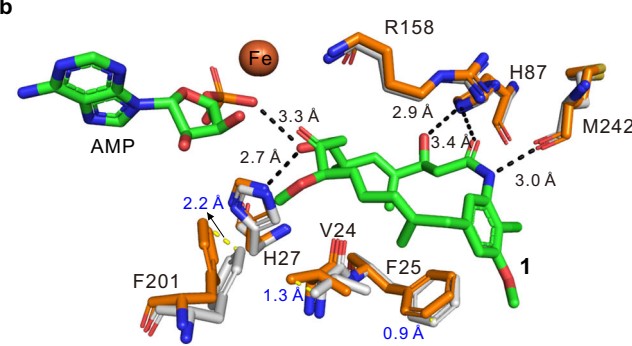

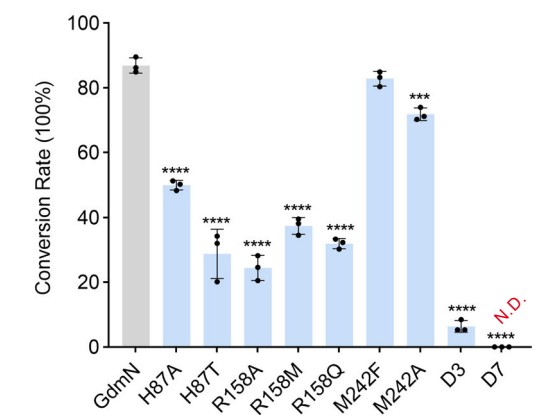

**Fig. 4 | The GdmN substrate-binding pocket and effects of deletions on catalytic activity. a** Binding position of **1** in the substrate-binding pocket. Orange, helix 1 (E85-E98), loop 1 (G8-A30), and loop 2 (D156-H162). Loop 4′ (H238-L246 from the other monomer) is not visible in this image. Cyan, helix 2 (L186-L196) and loop 3 (L197-D203). **b** Comparison of binding elements in the GdmN/AMP/**1** complex with the *apo* structure. Orange, residues in the GdmN/AMP/**1** complex; grey, residues in the *apo* structure. Hydrogen-bonding interactions are indicated with black dashed lines. **c** Catalytic activities of wild-type GdmN and its mutants. D3 and D7, mutants with deletion of I241-V243 and G239-N245, respectively. The activity assays of GdmN and mutants were performed with **1**, CP, ATP, and MgCl₂ at 30 °C for 12 h. N.D., products not detected. Wild-type GdmN was used as the control. Graphs depict means ± SD ($n = 3$ independent experiments). Statistical analysis was performed by one-way ANOVA with Dunnett's multiple-comparison test. ***$P < 0.001$ and ****$P < 0.0001$. Source data are provided as a Source Data file.

other known enzymes, suggesting a dimerization domain structure for a natural enzyme.

The N-terminal Kae1-like domain consisted of a two-stranded β-sheet and two five-strand β-sheets surrounded by eleven α-helices, containing a single iron ion coordinated by H133, H137, D156, and D333 in a square pyramidal geometry (Supplementary Fig. 10). The two histidines and two aspartates are highly conserved across class III

CTases (Supplementary Fig. 4). The binding pocket was composed of two helixes and four loops (Supplementary Fig. 11), i.e., helix 1 (E85-E98), helix 2 (L186-L196), loop 1 (G8-A30), loop 2 (D156-H162), loop 3 (L197-D203), as well as loop 4′ (H238-L246 from the other monomer), delineating a larger and more hydrophobic binding pocket than that of TobZ to accommodate ansamycin derivatives (Supplementary Figs. 12, 13, and 14). The face-to-face homodimer architecture further formed a reaction chamber inaccessible to solvent (Supplementary Fig. 12), favoring substrate transfer along a polarity gradient from the protein surface to the binding pocket and then stabilization at the active center. The large substrate pocket formed by the homodimer architecture indicated the potential for rational enzyme redesign to change the ansamitocin derivatives binding pose.

## Co-crystal structures of the GdmN complexes identified the substrate-binding elements

For the rational redesign of the substrate-specific enzyme into an iterative biocatalyst, it was necessary to understand the substrate-binding mechanism of GdmN. Therefore, the co-crystal structures of GdmN/AMP/**1** (2.80 Å resolution), GdmN/ATP (2.00 Å resolution), and GdmN/carbamoyl-AMP (1.98 Å resolution) were determined. The electron density clearly revealed that **1**, CP, carbamoyl-AMP, and the nucleotides were bound in different substrate-binding sites (Supplementary Figs. 15 and 16).

As observed in the GdmN/AMP/**1** structure, **1** with defined electron density was enveloped completely into each Kae1-like domain without direct access to bulky solvent. **1** was intimately held in proximity to the protein surface comprised of helix 1 (E85-E98), loop 1 (G8-A30), loop 2 (D156-H162), and loop 4′ (H238-L246 from the other monomer), distant from the other side of the binding pocket consisting of helix 2 (L186-L196) and loop 3 (L197-D203), and leading to an extra space in the binding pocket when GdmN bound **1** (Fig. 4a). This special binding mode indicated that, through appropriate engineering, GdmN might acquire an expanded scope for substrate recognition and bind **3** with the C-3 hydroxy group inserting into the active center, resulting in 3-*O* carbamoylation.

Closer inspection of **1** revealed the binding interactions with surrounding residues. The regionalization of hydrophobic residues and polar residues (H27, H87, and R158) oriented the binding pose of **1**, with the C-3 hydroxy group pointing to H87 and the hydrophobic side towards the hydrophobic surface consisting of V24, F25, F26, and F201 (Fig. 4b). Superimposition of the *apo* structure with the GdmN/AMP/**1** complex showed that the movements of the side chains of V24 and F25 resulted in better shape complementarity to accommodate **1**. The C-9 methoxy group of **1** formed a CH-π interaction with the benzene ring of F201, and the side chain of H87 formed hydrogen-bonding interactions with the C-3 hydroxy group and C-1 carbonyl group. Substituting H87 with Ala or Thr decreased the catalytic activity to 60% or 37% of the wild-type (Fig. 4c), implying that the hydrogen-bonding interaction provided by H87 plays an auxiliary role in **1** binding. The long side chain of R158 was in parallel with the orientation of the substrate, in which the guanidyl group formed a π-π interaction with the C-1 carbonyl group of **1**. Indeed, replacing R158 with Gln, Met, or Ala decreased the activity to 39%, 45%, and 33% of the wild-type, respectively (Fig. 4c). These results revealed that the π-π interaction between R158 and **1** also has a partial role in substrate binding. Moreover, the orientation of the side chain of R158 acts as a wall to secure the binding of **1**. The substitution of R158 with a shorter side chain might provide some redundant spatial environment to aid the binding of **3**.

Most notably, the backbone carbonyl of M242′ in loop 4′ formed hydrogen-bonding interactions with the amide of **1**, indicating that loop 4′ also participates in substrate binding. Replacing M242′ with other residues (Ala/Phe) did not affect the activity, whereas deletion of different amounts of residues in other mutants (mutants D3 and D7)

greatly decreased and even abolished carbamoylation ability (Fig. 4c), emphasizing the importance of loop 4′ in the homodimer architecture in terms of **1** binding. The interactions above, especially indispensable loop 4′, oriented **1** so that the C-7 hydroxy group is placed in the closest proximity to the imidazole ring of H27 at a distance of 2.7 or 2.5 Å, revealing that the general base H27 might act as proton receptor in the subsequent reaction.

In the GdmN/AMP/**1**, GdmN/ATP, and GdmN/carbamoyl-AMP complexes, the electron density of AMP, ATP, or carbamoyl-AMP only appears in the subdomain of the Kae1-like domain (Supplementary Figs. 15 and 16), and these compounds make similar contacts with surrounding residues to those with TobZ. Further experiments indicated that GdmN shares a similar adenylation mechanism with TobZ (Supplementary Fig. 17).

These results revealed the substrate-binding mechanism of GdmN for ansamitocin derivatives, in particular loop 4′ from the partner monomer playing a pivotal role for binding, further manifesting that the homodimer architecture not only provides a hydrophobic binding environment, but also provides the key elements for substrate binding. This subtrate-binding pocket with its regionalization of hydrophobic residues and polar residues differs from the binding pocket in TobZ, which has a mainly hydrogen-bonding network (Supplementary Fig. 14), revealing the natural specificity and co-evolution between substrates and binding pockets of different CTases. Notably, the position of **1** in the spacious binding pocket implied the suitability of this subclade of enzymes for reengineering into an iterative enzyme.

### Co-crystal structure snapshots characterized the active site center

To gain more insights into structural changes in the active site center during the reaction and the relation between transfer of the carbamoyl group and formation of the C7,9-cyclic carbinolamide group (Fig. 5a), we carried out multiple co-crystallization and soaking experiments by incubating GdmN with nucleotides, CP, and **1** at different time points. After several attempts, multiple crystal structures were determined. Unexpectedly, structural characterization revealed the appearance of three reaction intermediates at different stages, in which two different intermediates usually resided in different subunits of a single homodimer enzyme molecule (Supplementary Table 2, Supplementary Figs. 18 and 19). Such conditions indicated that reactions proceeded dependently in different subunits of homodimeric CTases, which has also been observed in other enzyme crystals[31]. One of the three intermediates was the natural tetrahedral intermediate carbamoyl-AMP-**1**, which had been predicted, but not structurally characterized, in a CTase. Another intermediate was **2**, which was carbamoylated, but not cyclized, with AMP located adjacent to **2**. The last intermediate was **1** together with carbamoyl-AMP in the subunit. The above observations provided the descriptions of the atomic coordinates of multiple intermediates for a CTase without the use of protein mutants.

In the subunit containing **1** and carbamoyl-AMP (Fig. 5b, Supplementary Figs. 20b and 21), the C-7 oxygen atom of **1** was 2.8 Å from the imidazole ring of H27 and 2.8 Å from the carboxamide carbon of carbamoyl-AMP, indicating that H27 is inevitably protonated and subsequently initiates the reaction to drive deprotonated **1** to engage in nucleophilic attack at the carbamoyl group. Carbamoyl-AMP appears to surround the iron ion binding site, with the amine of the carbamoyl group coordinated to iron ion and the carbonyl oxygen accommodated by the conventional oxyanion hole composed of the backbone amides of R158 and G157[32].

Structural analysis of the carbamoyl-AMP-**1** tetrahedral intermediate revealed the geometric structure of a natural tetrahedral intermediate in a carbamoylation process (Fig. 5c, Supplementary Figs. 20c and 21). The carbamoyl-AMP portion remained almost in the same position, while the formation of the C-O bond between the carbamoyl group and the oxygen atom of C-7 in **1** drove a slight shift of **1**

towards carbamoyl-AMP (Fig. 5e and Supplementary Fig. 21). The capture of the tetrahedral intermediate implied that this catalytic reaction proceeds in a stepwise manner rather than in a concerted process.

In the GdmN/AMP/**2** unit, the C-7 oxygen of **2** was 2.9 Å from the lateral chain of H27 (Fig. 5d, Supplementary Figs. 20d and 21). The distance between the amine of the carbamoyl group and iron ion was 3.1 Å, while the distance was 2.7 Å in the GdmN/carbamoyl-AMP/**1** subunit, representing the end of the carbamoylation reaction. The structural characterization of **2** revealed the conformation of ansamycin derivatives before the formation of the C7,9-cyclic carbinolamide group, which has not been observed in other studies (Fig. 5f). Based on the geometric structure of **2**, cluster-continuum model calculations were performed[33,34]. The results showed that the conversion from **2** to **3** in the presence of water molecules involves an overall Gibbs free energy barrier of 15.3 kcal/mol, revealing that water molecules would deprotonate the amino group of the carbamoyl group from **2**, leading to nucleophilic attack on C-9 of **2** to generate **3** (Supplementary Figs. 22 and 23).

Structural insights revealed that H27 plays a deprotonation role during reactions (Supplementary Fig. 21). Further mutation of H27 to the neutral hydrogen bond donor Asn caused the complete loss of activity (Fig. 5g), confirming that the positively charged imidazolium group is essential for catalysis. The histidine was universally conserved in this CTase subclade (Supplementary Fig. 4), suggesting that these CTases utilize the histidine to initiate the carbamoylation reaction. Intriguingly, a water molecule (W′) interacting with the hydroxy group of Y82 appears near H27 in all experimental data (Supplementary Fig. 21), forming hydrogen-bonding interactions with the backbone amide and carbonyl of H27. Y82 was highly conserved in this subclade (Supplementary Fig. 4), and replacing Y82 with Phe abolished the activity (Fig. 5g), revealing the importance of Y82 in making water-mediated (W′) hydrogen bonds. We speculated that the W′ may maintain the conformation of loop 1 or participate in proton delivery during the reaction. To interpret the role of Y82, we determined the structure of mutant Y82F (2.30 Å resolution) and found it similar to that of *apo* GdmN, with the major difference occurring in loop 1 (Supplementary Fig. 24). In Y82F, a substantial structural swing of loop 1 caused the side chain of H27 to rotate almost 125°, far away from the active center. Superimposition of the Y82F structure with that of the GdmN/AMP/**1** complex revealed that loop 1 protruded into the binding pocket, causing steric clashes with **1**, thereby leading to complete elimination of activity (Fig. 5h).

Thus, our study revealed that Y82 is the keystone residue in the binding pocket in this CTase subclade. The multiple structural snapshots depicted the active site center during the reaction. To get more insights into the catalytic mechanism, QM/MM calculations on the enzymatic reactions were performed (Supplementary Fig. 25). The results showed that the enzymatic reaction from **1** to **2** requires a barrier of 17.7 kcal/mol, which is higher than the barrier of the conversion from **2** to **3** in water solution. These results implied that the first step of 7-*O* carbamoylation reaction should be the rate-limiting step.

### Structure-guided engineering enabled GdmN to iteratively install carbamoyl groups onto 1 at the C7 and C3 hydroxy groups

To generate a di-*O*-CTase based on our structural analyses (Fig. 6a), several issues were considered. Firstly, such a transformation required that **1** be preferentially stabilized at appropriate positions to allow 7-*O*-carbamoylation prior to 3-*O*-carbamoylation, thus avoiding changes in structural elements vital for orientation of **1**, such as loop 4′ and H87. Secondly, appropriate replacement of residues in the binding pocket was likely to remodel enzyme regioselectivity, as reported in other studies[35–37]. Subsequently, the docking studies using the GdmN/carbamoyl-AMP complex and **3** suggested that GdmN would be unable to

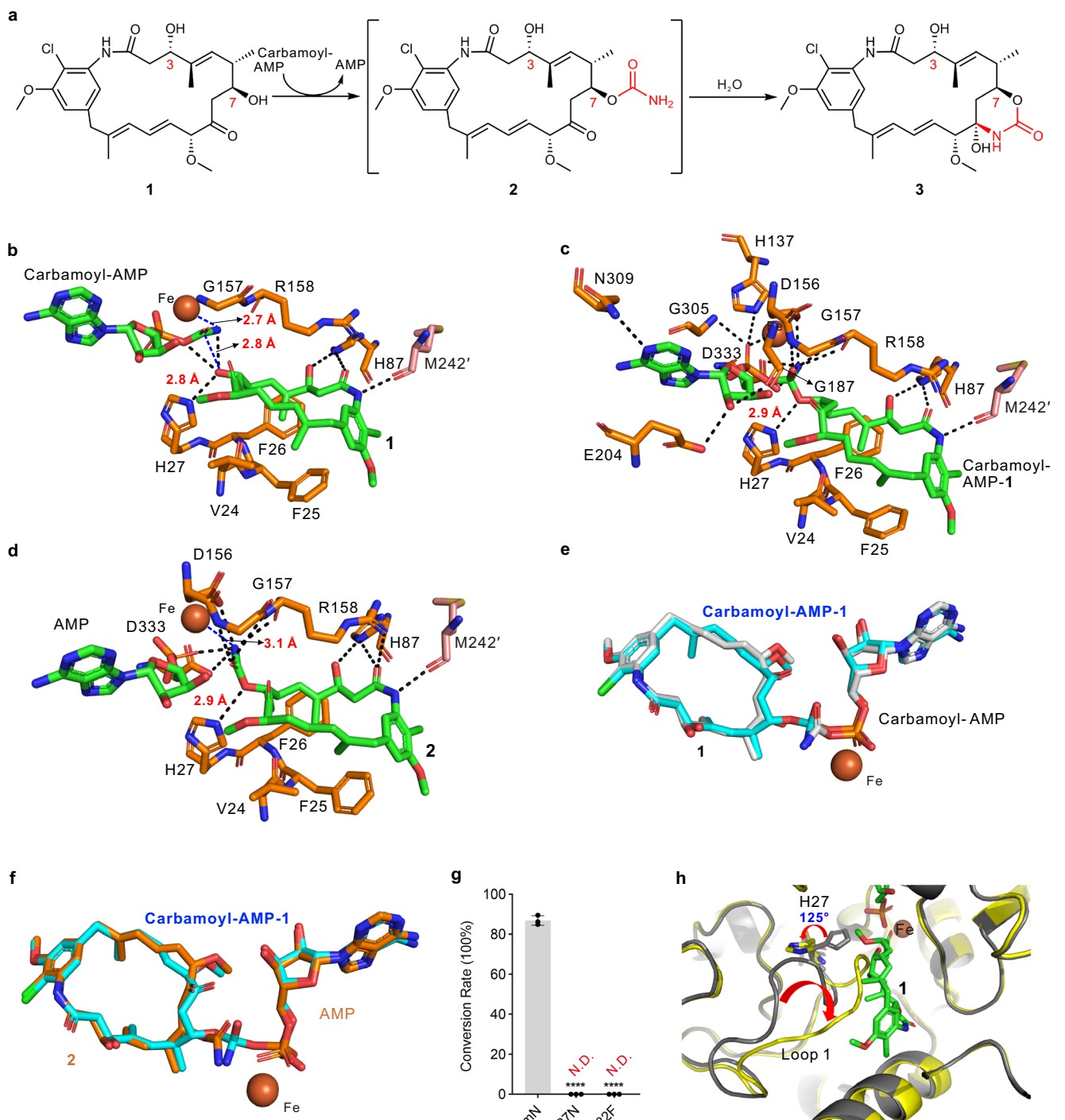

**Fig. 5 | Mechanistic insights into the GdmN-catalyzed reaction based on multiple structural snapshots. a** The proposed reaction involved in carbamoylation and formation of the C7,9-cyclic carbinolamide group. **b, c, d** Interaction of ligands with surrounding residues in the GdmN/carbamoyl-AMP/**1**, GdmN/carbamoyl-AMP-**1**, and GdmN/AMP/**2** structures, respectively. The residues are shown as orange stick models. M242′ shown as pink stick model is located in loop 4′ from the partner monomer. Hydrogen-bonding interactions are indicated with black dashed lines. **e** Superimposition of carbamoyl-AMP-**1** onto carbamoyl-AMP. Grey stick models, structures of carbamoyl-AMP and **1**; cyan stick model, structure of carbamoyl-AMP-**1**. **f** Superimposition of carbamoyl-AMP-**1** onto **2** and AMP. Orange stick model, structures of AMP and **2**; cyan stick model, structure of carbamoyl-AMP-**1**. **g** Catalytic activities of wild-type GdmN, H27N, and Y82F. N.D., products not detected. The activity assays of GdmN and mutants were performed with **1**, CP, ATP, and $MgCl_2$ at 30 °C for 12 h. Wild-type GdmN was used as the control. Graphs depict means ± SD ($n = 3$ independent experiments). Statistical analysis was performed by one-way ANOVA with Dunnett's multiple-comparison test (****$P < 0.0001$). Source data are provided as a Source Data file. **h** Superimposition of the Y82F structure (yellow) onto the GdmN/AMP/**1** structure (grey).

bind **3** in the expected orientation, as we assumed that **3** might have steric interference with loop 2 (D156-H162). In addition, the enzyme could not provide potentially sufficient interactions to bind **3** due to the large unoccupied space appearing in the vicinity of loop 1 (G8-A30)

and helix 2 (L186-L196) in the active pocket of the GdmN/carbamoyl-AMP/**1** complex.

Due to the 61% protein sequence identity of GdmN with Asc21b responsible for 3-*O*-carbamoylation, homology modelling of Asc21b

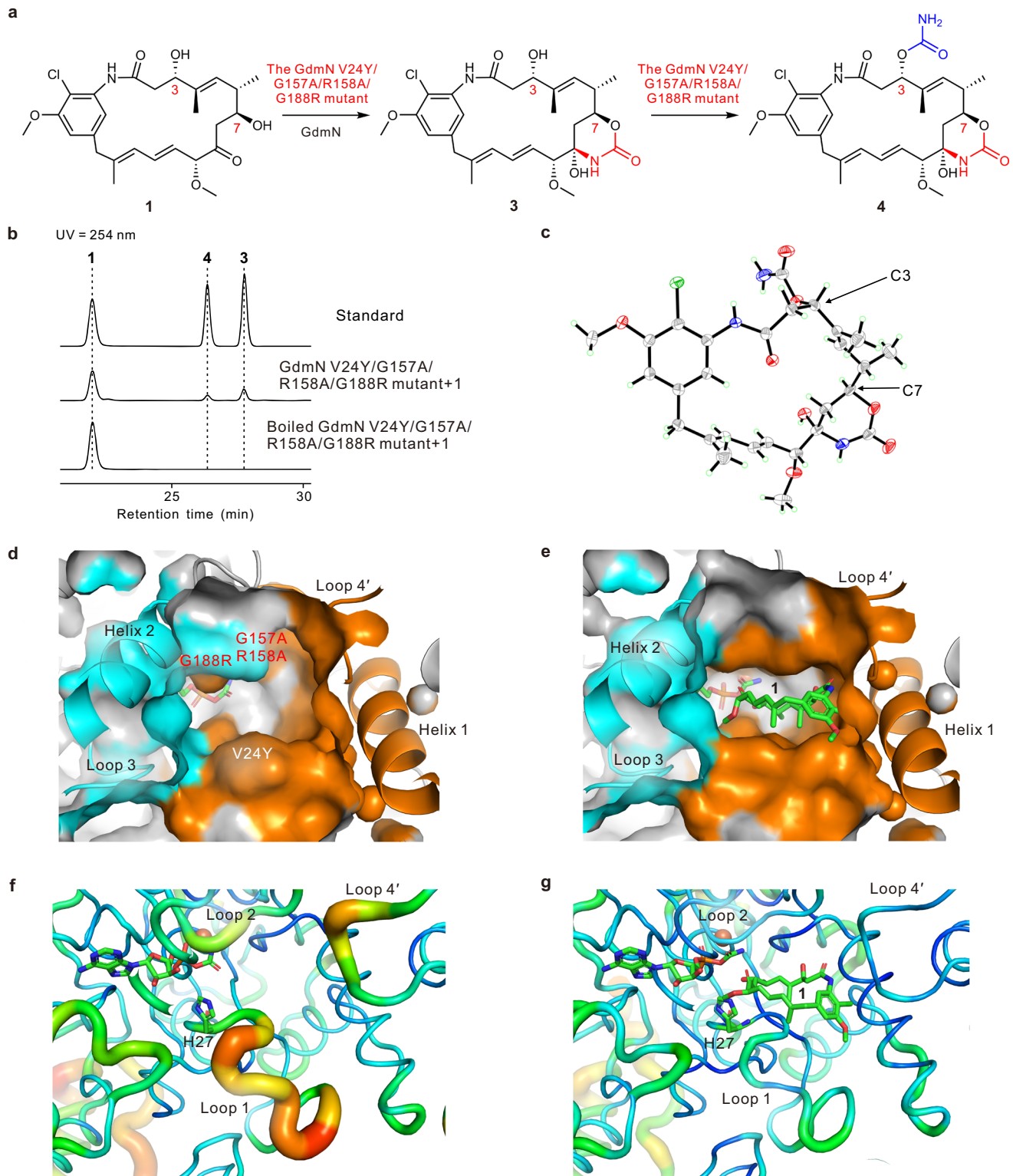

**Fig. 6 | Analysis of the iterative mutant of GdmN. a** The di-*O*-carbamoylation reaction catalyzed by the GdmN V24Y/G157A/R158A/G188R mutant. **b** HPLC results showing conversion of **1** to **3** and **4** by the mutant. The activity assays of the mutant were performed with **1**, CP, ATP, and MgCl₂ at 30 °C for 24 h. **c** ORTEP structure for **4** (50% probability ellipsoids; water molecules are omitted for clarity). **d, e** The substrate binding pocket of the GdmN V24Y/G157A/R158A/G188R mutant/ carbamoyl-AMP complex and the GdmN/carbamoyl-AMP/**1** complex, respectively. Orange, helix 1 (E85-E98), loop 1 (G8-A30), loop 2 (D156-H162), and loop 4′ (H238-L246 from the other monomer is not visible in these images); cyan, helix 2 (L186-L196) and loop 3 (L197-D203). **f, g** b-factor putty views of the mutant/carbamoyl-AMP complex and the GdmN/carbamoyl-AMP/**1** complex, respectively.

based on GdmN was performed via the SWISS-MODEL server (Supplementary Fig. 26)[38]. We then examined differences in GdmN and Asc21b structural elements and considered five residues that might influence substrate orientation (Supplementary Fig. 27). Further analysis with structural characterization of loop 1 (G8-A30) and helix 2 (L186-L196) suggested that G188, one of the residues lining the surface of the active site in GdmN, is mutated to another bulky residue, Arg, in Asc21b. The homology model of the mutant G188R indicated that the side chain of the Arg residue could protrude into the binding pocket, which may donate hydrogen bonds to facilitate binding of **3**. A similar condition also occurred with V24. We presumed that the substitution of Tyr for Val could offer necessary interactions for **3** binding. To remove a possible steric clash with **3**, we speculated that replacing G157 and R158 in GdmN with Ala might enable the desired function. Due to the hydrogen-bonding role of loop 4′ and H87 participating in the orientation of **1**, these sequences were not changed to reduce structural perturbation that might interfere with the 7-*O*-carbamoylation function.

To reengineer GdmN as an iterative catalyst, we constructed a series of site-directed mutants by reciprocally exchanging the above-mentioned residues and incubated these mutants with **1** or **3** in the presence of $Mg^{2+}$, ATP, and CP (Supplementary Fig. 28). Catalytic activity results revealed that the point mutants were able to catalyze C-7 carbamoylation of **1** with decreased catalytic efficiency and that they could not catalyze the conversion of **3** to other compounds, indicating that simple point mutations could not change regioselectivity. As multiple mutations might appropriately reshape the binding pocket, allowing **3** to bind, multi-site mutants were constructed and incubated with **1** or **3** in the presence of $Mg^{2+}$, ATP, and CP. All multi-site mutants converted **1** into **3**, whereas only the GdmN-V24Y/G157A/R158A/G188R mutant could catalyze the conversion of **3** to a new compound, namely **4** (HR-ESI-MS *m/z* 600.2092 [*M* + Na]+), albeit with relatively low catalytic efficiency (Fig. 6a, b, and Supplementary Fig. 28). The mass difference between **3** and **4** indicates that **4** is a carbamoylated derivative of **3**. The structure of **4** was determined by NMR spectroscopy and HR-ESI-MS (Supplementary Figs. 29 and 30) and confirmed by X-ray crystallographic analysis (Fig. 6c and Supplementary Table 3), demonstrating that this mutant could catalyze 7-*O* carbamoylation and 3-*O* carbamoylation successively.

The crystal structure of the mutant with carbamoyl-AMP was determined and was found to be similar to that of the GdmN/carbamoyl-AMP/**1** complex, with an r.m.s.d. value of 0.26 Å (Supplementary Fig. 31). The major difference occurred at the substrate-binding pocket (Supplementary Fig. 32). As we had hypothesized, G188R and V24Y reshaped the substrate-binding pocket, refining the shape complementarity of the binding pocket to the substrates (Fig. 6d, e, and Supplementary Fig. 33). The smaller side chain resulting from the R158A mutation might offer more space and reduce the spatial hindrance of **3**. Due to the G157A change, the carbonyl oxygen atom of the carbamoyl group could not be accommodated by the conventional oxyanion hole composed of the backbone amides of A158 and A157, leading to a rotation of the carbamoyl group of carbamoyl-AMP (Supplementary Figs. 34 and 35). Additional b-factor putty views indicated that P19-F25 of loop 1, D156-H162 of loop 2, and N240-P244 of loop 4′ in the iterative mutant are highly mobile compared to those in wild-type GdmN (Fig. 6f, g). Such flexible loops resulted in more possibility for **3** binding.

To investigate the binding mechanism for **3** of the mutant, we conducted multiple attempts but were unable to obtain such a complex. Thus, the computational docking of **3** with the mutant structure was performed (Supplementary Fig. 36). **3** bound to the mutant in a direction in which the C-3 oxygen was placed close to the carbamoyl-AMP carbamoyl group and the side chain of His27 at distances of 3.6 and 4.4 Å, respectively, explaining the occurrence of 3-*O* carbamoylation. Other residues lining the binding pocket underwent varying degrees of fluctuation to bind **3**. The amide of **3** formed a hydrogen-bonding interaction with the backbone carbonyl of A158, and the flexible side chain of R188 donated a hydrogen bond with the C-7 oxygen. Y24, together with F25 and F26, engaged in hydrophobic interactions with the nonpolar side of **3**. 20-*O*-methyl interacted with Y82 and F83 via the CH-π interactions. To further elucidate the occurrence of 3-*O* carbamoylation catalyzed by the mutant, we have performed MD simulations (Supplementary Figs. 37 and 38). The results showed that C-3 oxygen of **3** and the carbamoyl-AMP carbamoyl group maintain a favorable distance for 3-*O* carbamoylation in the MD simulations of the mutant, while the distance in the MD simulations of GdmN is unfavorable for 3-*O* carbamoylation. All the calculations are consistent with our experiments results.

Overall, structure-based site-specific replacement of multiple plasticity residues lining the binding pocket reprogrammed the regioselectivity of GdmN. Further crystallographic studies disentangled the effects of amino acid replacement at multiple sites, and computational analysis unveiled the occurrence of 3-*O* carbamoylation of **3**, providing insights into this unprecedented example of repurposing a canonical CTase into an iterative enzyme. Moreover, these results inspired us to gain more insights into other CTases in this subclade via homology modeling (Supplementary Fig. 39), revealing that the multiple plasticity residues, which differ in this subclade of CTases, are the consequence of natural evolution and account for functional innovation in the product and substrate spectrum of these CTases.

## Discussion

Whereas canonical enzymes have strict specificity for their substrates, iterative enzymes catalyze two or more modification steps, providing greater potential for many synthetic applications[39]. However, high-throughput screening for naturally iterative enzymes is time-consuming and labor-intensive. Therefore, redesigning enzymes to perform specific iterative functions has become an attractive, although challenging, alternative. However, directed evolution to redesign enzymes through mutagenesis can lead to unpredictable deleterious effects on function or stability and requires the evaluation of large mutant libraries. Reengineering strategies driven by structural and mechanistic insights provide a more practical approach for generating the desired enzymes[21]. Hence, we characterized multiple crystal structures of GdmN in complex with different intermediates and performed functional studies through mutagenesis experiments, providing key information on the structural and mechanistic features of homodimeric CTases. Protein homology modeling suggests overall similarities in the binding pocket in this subclade, but with subtle alterations in plasticity residues that contribute to the considerable substrate flexibility of these CTases (Supplementary Fig. 39). The alternation lining binding pocket inspires us to reprogram the regioselectivity of GdmN by exchanging just a few amino acids. The point mutants could not change the product distributions, while multi-site mutagenesis could generate an expected iterative CTase for the one-pot synthesis of maytansinoids derivatives with successive 7-*O* and 3-*O* carbamoylations (Fig. 6b and Supplementary Fig. 28).

The crystallographic studies provided experimental support for the overall mechanism, which had not previously been achieved[16]. It was shown that carbamoyl-AMP and substrates enter reactive position respectively, and then forms a tetrahedral intermediate assisted by H27 (Supplementary Fig. 21). Computational studies implied that the formation of the tetrahedral intermediate is the rate-limiting step (Supplementary Figs. 22 and 25). Subsequently, the decomposition of the intermediate generates carbamoylated product. Further QM calculations also elucidated formation of the C7,9-cyclic carbinolamide group during ansamycin biosynthesis (Supplementary Fig. 22). Moreover, the present study also observed the interesting result that the conserved tyrosine (Y82 in GdmN and Y82 in NovN) plays a key role in the binding pocket of this subclade (Fig. 5h).

Characterization of the homodimeric CTases put the last piece of the puzzle of known CTases, providing insights into a possible evolutionary relationship between CTases and their substrates. Class III CTases catalyze formation of an extensive variety of secondary metabolites with chemical diversity (Supplementary Fig. 1) and differing from class I and class II CTase substrates (Supplementary Fig. 2). The class III CTases that lack a dimerization domain usually catalyze modification of polar recipients, such as tobramycin for TobZ[16] and streptothrisamines for StnQ[40]. Consistently, these CTases have a binding pocket exposed to solvent and function as a monomer, binding the substrate through hydrogen-bonding interactions[16]. By contrast, CTases in the GdmN subclade favor weak polar and macrocyclic compounds[11,24,25,41], and their relatively hydrophobic binding pockets are further stabilized by the homodimeric architecture. Under natural pressure to produce these carbamoylated compounds, the ancestral GdmN-type CTases may have evolved a relatively hydrophobic binding pocket and subsequently developed the dimerization domain to enhance catalytic efficiency. These CTases may offer a model system for investigating the co-evolution between natural substrates and enzyme specificity.

Our study also further unveiled the recognition mechanism for iterative activity, revealing that the hydrogen-bonding interaction donated by G188R and more flexibility of the loops resulted from V24Y, G157A, and R158A, contributes collaboratively to the regioselectivity modulation (Supplementary Fig. 36). The loop engineering strategy has been applied in generation of novel biocatalysts recently[42,43]. Our study showed that the combination of loop engineering and appropriate variations lining active-site cavity will yield other intriguing results, which could be applied in further enzyme engineering.

Rational replacement of these plasticity residues should enable reprogramming of the regioselectivity of these CTases, as demonstrated by our current study in which replacement of residues at multiple sites had a synergistic effect on regioselectivity, thus providing a blueprint for multi-site protein engineering for targeted function. Our approach may aid in the rational engineering of other iterative enzymes, allowing the creation of catalytic tools for the efficient biosynthesis of antimicrobial and anticancer compounds.

## Methods
### General materials
DNA primers were synthesized by Sangon Biotech. All chemicals were purchased from Sigma-Aldrich, Sangon Biotech, and Thermo Fisher Scientific. Mass analysis was carried out on an Agilent High-Resolution Quadrupole Time of Flight (QTOF) 6545 MS with an electrospray ionization (ESI) source.

### Bacterial strains, plasmids and culture conditions
Strains and plasmids used in this study are listed in Supplementary Table 4. *Escherichia coli* DH10B was used for all plasmids construction, and *E. coli* BL21(DE3)pLysS was used for protein expression. *E. coli* ET12567(pUZ8002) was used for conjugation. *E. coli* strains were cultivated in Luria-Bertani (LB) broth with corresponding antibiotics. *Actinosynnema pretiosum* subsp. *pretiosum* ATCC 31280 was obtained from the ATCC Global Bioresource Center. *A. pretiosum* ATCC 31280 and the mutants were cultivated in TSBY liquid medium or on ISP2 agar plates[44]. The TSBY liquid medium contained tryptone soya broth (30 g l$^{-1}$), yeast extract (5 g l$^{-1}$), sucrose (103 g l$^{-1}$), pH 7.5. The ISP2 agar plate consisted of glucose (4 g l$^{-1}$), malt extract (10 g l$^{-1}$), yeast extract (4 g l$^{-1}$), pH 7.5, and agar (15.0 g l$^{-1}$). The IPS2 agar plates were used for the *E. coli-Actinosynnema* bi-parental conjugation.

For fermentation of ansamitocins derivatives, *A. pretiosum* ATCC 31280 and the mutants were firstly cultivated in liquid TSBY medium at 30 °C, 220 r.p.m. for 24 h. 3.3% inoculum was added into liquid seed medium (tryptone soya broth 30 g l$^{-1}$, yeast extract 8 g l$^{-1}$, sucrose 103 g l$^{-1}$, isopropanol 500 μl l$^{-1}$, isobutanol 500 μl l$^{-1}$, pH 7.5) and then cultivated at 30 °C, 220 r.p.m. for 24 h. Subsequently, 10% inoculum from the seed medium was added into liquid fermentation medium (malt extract 10 g l$^{-1}$, yeast extract 16 g l$^{-1}$, sucrose 103 g l$^{-1}$, isopropanol 12 ml l$^{-1}$, isobutanol 5 ml l$^{-1}$, MgCl$_2$ 2 mM, pH 7.5) at 25 °C, 220 r.p.m. for 7 days. Fermentation was conducted in baffled 250-ml flasks with 30 ml corresponding liquid medium.

### Gene inactivation and complementation
In the biosynthetic machinery of ansamitocin in *A. pretiosum* ATCC 31280 (Supplementary Fig. 5), gene *ansa23* encoding carbamoyltransferase and gene *ansa21* encoding acyltransferase were inactivated using homologous recombination technology. To construct plasmid for the inactivation of the *ansa23*, the genomic DNA of *A. pretiosum* ATCC 31280 was used as a template. The upstream and downstream homologous arms were amplified with *ansa23*-deletion-L-FP/RP primers and *ansa23*-deletion-R-FP/RP primers (Supplementary Table 5), respectively. Both arms were cloned into the *Bam*HI and *Hin*dIII sites of *Actinomycete–E. coli* shuttle vector pJTU1278 to afford plasmid pLQ1051. pLQ1051 was verified by sequencing and then introduced into *E. coli* ET12567(pUZ8002). The *E. coli-Actinosynnema* bi-parental conjugation was performed to introduce pLQ1051 into *A. pretiosum* ATCC 31280. After cultured for 5- day incubation at 30 °C, exconjugants grown on ISP2 plates with 20 mg ml$^{-1}$ thiostrepton were cultivated in liquid TSBY medium and verified by PCR with *ansa23*-deletion-verify-FP/RP primers. Then, the correct single-crossover strains were inoculated into TSBY without antibiotics and incubated for 24 h at 30 °C, to eliminate the plasmid. Thiostrepton-sensitive strains were further verified by PCR with *ansa23*-deletion-verify-FP/RP primers to generate the mutant WJH10 (Supplementary Fig. 40).

For the inactivation of gene *ansa21*, the upstream and downstream homologues arms were individually amplified with *ansa21*-deletion-L-FP/RP and *ansa21*-deletion-R-LP/RP (Supplementary Table 5). Both arms were cloned into the *Bam*HI and *Hin*dIII sites of pJTU1278 to obtain plasmid pLQ1052. The plasmid was induced into *E. coli* ET12567(pUZ8002) by transformation and further into *A. pretiosum* ATCC 31280 by conjugation. After 5-day incubation at 30 °C, exconjugants grown on thiostrepton plates. Then, the exconjugants were cultivated in TSBY and verified by PCR using *ansa21*-deletion-verify-FP/RP primers. The correct single-crossover strains were inoculated into TSBY without antibiotics and incubated for 24 h at 30 °C. The correct double-crossover strains were selected by loss of thiostrepton resistance and PCR verification to generate the mutant WJH13 (Supplementary Fig. 41).

For heterologous expression of *gdmN* in WJH10, the genomic DNA of *Streptomyces hygroscopicus* XM201 was used as the PCR template[45]. The *gdmN* gene was amplified with *gdmN*-FP/RP primers (Supplementary Table 5) and inserted into the *Nde*I and *Eco*RI sites of pLQ646 to obtain pLQ1053[44]. After DNA sequencing, the correct vectors were transferred into *E. coli* ET12567(pUZ8002) and then conjugated into WJH10. The correct recombinants were screened by apramycin resistance and PCR verification to generate the mutant WJH11 (Supplementary Fig. 42).

To constitutively overexpress the truncated *gdmN* gene (M1-L575) in WJH10, the truncated *gdmN* gene (M1-L575) was amplified with GdmNm (mutant of C-terminal deletion)-FP/RP primers (Supplementary Table 5) and cloned into the *Nde*I and *Eco*RI sites of pLQ646 to afford pLQ1054. The plasmid pLQ1054 was conjugated into WJH10 to generate the derivative WJH12 (Supplementary Fig. 43).

### Gene expression and protein purification
The GdmN coding sequence was amplified by PCR using *gdmN*-FP/RP primers (Supplementary Table 5) with genomic DNA of *S. hygroscopicus* XM201 as the template. The PCR fragment was cloned into the pET-28a(+) digested by *Nde*I and *Eco*RI, to obtain pLQ1055. After DNA

sequencing, the plasmid pLQ1055 was introduced into *E. coli* BL21(DE3)pLysS for heterologous expression. The single colony was cultivated overnight at 37 °C in 50 ml of LB medium containing 50 μg ml⁻¹ kanamycin and 34 μg ml⁻¹ chloramphenicol. 50 ml overnight culture was transferred to 5 l LB medium containing 50 μg ml⁻¹ kanamycin and 34 μg ml⁻¹ chloramphenicol. Then, the culture was cultivated at 37 °C, 220 r.p.m. until the $OD_{600}$ value reached 0.8. The culture was cooled to 4 °C and then induced with 0.4 mM IPTG. The culture was further cultivated at 16 °C and 220 r.p.m. for 18 h. The cells were harvested and resuspended in lysis buffer (50 mM Tris, 500 mM NaCl, pH 8.0). The cells were centrifuged after ultrasonication, and the supernatant was purified using gravity chromatography columns with Ni Sepharose 6 Fast Flow beads (Cytiva) at 4 °C. The column was washed with washing buffer A (50 mM Tris, 500 mM NaCl, 50 mM imidazole, pH 8.0), and the protein was eluted with elution buffer (50 mM Tris, 500 mM NaCl, 300 mM imidazole, pH 8.0). Fractions containing GdmN were further purified by a fast protein liquid chromatography system (Bio-Rad) with a Superdex 200 Increase 10/300 GL gel filtration column (Cytiva) pre-equilibrated in buffer C (10 mM Tris, 150 mM NaCl, pH 8.0). The fractions of the highest purity as evaluated by SDS-PAGE (Supplementary Fig. 6) were concentrated to 10 mg ml⁻¹.

### Site-directed mutagenesis of GdmN

The primers listed in Supplementary Table 5 were used for construction of the mutants, and the *gdmN*-containing plasmid pLQ1055 was used as the template. The purified PCR products were incubated with *Dpn*I and T4 DNA ligase, and then introduced into BL21(DE3)pLysS. Each mutation was verified by DNA sequencing. GdmN mutants were purified as described above for native protein purification.

### Protein crystallization procedure

For GdmN (PDB: 7VYO) and GdmN Y82F mutant (PDB: 7VZU), initial screenings were performed with crystallization screen kits (Hampton Research) by sitting-drop vapor-diffusion method at 20 °C. 1 μl drop consisted a 1:1 mixture of purified protein solutions (10 mg ml⁻¹) and precipitant solutions. Needle crystals were observed in a precipitant solution (0.2 M lithium sulfate monohydrate, 25% (w/v) PEG3350 and 0.1 M Tris, pH 8.5) at 20 °C. After the optimization of pH and precipitant, rhomboid crystals of GdmN were obtained in a precipitant solution containing 0.2 M lithium sulfate monohydrate, 30% (w/v) PEG3350 and 0.1 M Tris, pH 9.0, and rhomboid crystals of GdmN Y82F mutant were obtained in a crystallization buffer containing 0.2 M lithium sulfate monohydrate, 27.5% (w/v) PEG3350 and 0.1 M Tris, pH 9.0.

For the complex of GdmN with ATP (PDB: 7VX0), purified GdmN solution (10 mg ml⁻¹) was incubated with 1.5 mM ATP for 1 h on ice. The 1 μl protein solution was then mixed with 1 μl precipitant solution. Rhomboid crystals were obtained in a precipitant solution containing 0.2 M lithium sulfate monohydrate, 28% (w/v) PEG3350 and 0.1 M Tris, pH 9.0.

For the complex of GdmN with AMP and **1** (PDB: 7VZY), purified GdmN solution (10 mg ml⁻¹) was incubated with 1.5 mM CP, 1.5 mM AMP, and 1 mM **1** for 1 h on ice. The 1 μl protein solution was then mixed with 1 μl precipitant solution. Rhomboid crystals were obtained in a precipitant solution containing 0.2 M lithium sulfate monohydrate, 30% (w/v) PEG3350 and 0.1 M Tris, pH 9.0.

For the complex of GdmN with carbamoyl-AMP (PDB: 7VYJ), purified GdmN solution (10 mg ml⁻¹) was incubated with 1.5 mM ATP, 1.5 mM MgCl₂, and 1.5 mM CP for 1 h on ice. The 1 μl protein solution was then mixed with 1 μl precipitant solution. Rhomboid crystals were obtained in a precipitant solution containing 0.2 M lithium sulfate monohydrate, 32.5% (w/v) PEG3350 and 0.1 M Tris, pH 8.5.

For the complex of GdmN V24Y/G157A/R158A/G188R mutant with carbamoyl-AMP (PDB: 7VZQ), purified GdmN mutant solution (10 mg ml⁻¹) was incubated with 1.5 mM ATP, 1.5 mM MgCl₂, and 1.5 mM CP for 1 h on ice. The 1 μl protein solution was then mixed with 1 μl precipitant solution. Rhomboid crystals were obtained in a precipitant solution containing 0.2 M lithium sulfate monohydrate, 27.5% (w/v) PEG3350 and 0.1 M Tris, pH 9.0.

For the complex of GdmN with natural tetrahedral intermediates, carbamoyl-AMP and **1** (PDB: 7VZZ) and the complex of GdmN with natural tetrahedral intermediates, AMP, and **2** (PDB: 7VYP), purified GdmN solution (10 mg ml⁻¹) was incubated with 1.5 mM ATP, 1.5 mM MgCl₂, 1.5 mM CP and 1 mM **1** for 2 h on ice. 1 μl protein solution was then mixed with 1 μl precipitant solution. After 2 days, rhomboid crystals were obtained in a precipitant solution containing 0.2 M lithium sulfate monohydrate, 30.0% (w/v) PEG3350 and 0.1 M Tris, pH 9.0.

For the complex of GdmN with **1** and carbamoyl-AMP (PDB: 7VZN), multiple crystals of GdmN with carbamoyl-AMP were soaked in a precipitant solution containing 0.2 M lithium sulfate monohydrate, 30.0% (w/v) PEG3350, 0.1 M Tris, pH 9.0, and different concentrations of **1**.

All crystals were soaked in a crystallization solution containing 20% (v/v) glycerol before being placed into liquid nitrogen.

### Structural determination and refinement procedure

All X-ray diffraction data of GdmN crystals were collected at the Beam Line 18U1 and 19U1 at the Shanghai Synchrotron Radiation Facility (SSRF). All datasets of different crystals were indexed, integrated, and scaled using the HKL2000 or HKL3000 package[46,47]. Structure of GdmN was determined by molecular replacement using the program Phaser 2.7.17 from Phenix suite[48], whereby the structure of TobZ (PDB: 3ven) was used as a model. Iterative cycles of model rebuilding and refinement were performed using COOT[49] and Phenix, to generate the final model of GdmN. Structures of other GdmN complexes were determined by molecular replacement using Phenix and the atomic coordinates of GdmN as a model. Refinement with Phenix[48] along with model adjustment and ligand placement with COOT[49] generated final models of GdmN complexes. MolProbity from Phenix suite was used to evaluate the quality of the structural models. The Refinement statistics of different structures are summarized in Supplementary Table 2. Structure figures were prepared with PyMOL (The PyMOL Molecular Graphics System, Version 2.3.1 Schrödinger, LLC).

### In vitro biochemical assays

The reaction mixture (100 μl) for the biochemical assays of GdmN and mutants, consisting of 10 mM MgCl₂, 1 mM carbamoyl phosphate, 2 mM ATP, 0.5 mM **1** or **3**, 50 mM Tris, pH 8.5, 200 mM NaCl, and 40 μM proteins, was incubated at 30 °C for 12 h. Heat-inactivated GdmN (boiling at 100 °C for 15 min) was used as the negative control. The reactions were quenched with 100 μL methanol. Precipitated proteins were removed by centrifugation, and the supernatants were subjected to HPLC analysis. HPLC analyses were performed on the Agilent 1290 infinity instrument with an Agilent Eclipse Plus-C18 analytic column (4.6 × 150 mm, 5-μm particle size) equilibrated with 90% solvent A (water) and 10% solvent B (methanol). A gradient of 10–50% methanol was applied from 0 to 5 min, followed by the following gradients: 50%-60% methanol from 5 to 20 min, 60%-75% methanol from 20 to 28 min, 75%-95% methanol from 28 to 36 min, 95% methanol from 36 to 45 min, 95%-10% methanol from 45 to 46 min, and re-equilibration in 10% methanol from 46 to 55 min. The flow rate was 0.5 ml min⁻¹, and absorbance was measured at 254 nm. The reactions were performed in triplicate ($n = 3$). The activity assays were analyzed using GraphPad Prism version 9.0.2.

### Phylogenetic analysis

Amino acid sequences of different CTases were aligned by ClustalW multiple alignments[50]. The alignment result was analyzed by the maximum likelihood method using the Molecular Evolutionary Genetics Analysis (MEGA 11) software[51]. The phylogenetic analysis used

bootstrap test (1000 replicates) to check the robustness of result. The phylogenetic tree was further decorated by Interactive Tree of Life (iTOL)[52].

## System setup and MD simulations

Three systems have been setup for the MD simulations: (a) GdmN/carbamoyl-AMP/**1**; (b) GdmN/carbamoyl-AMP/**3**; (c) GdmN V24Y/G157A/R158A/G188R mutant/carbamoyl-AMP/**3**. As GdmN formed a face-to-face homodimer through unusual C-terminal domains, the interactions between chain A and chain B should be taken into consideration. So, both chain A and chain B were retained in our MD simulations. The protonation states of titratable residues (His, Glu, Asp) were assigned on the basis of pKa values by PROPKA[53] and visual inspection of local H-bonding networks. Histidine residues 27, 87, 133, 137, 238, 297, 308, and 422 were protonated at the δ position. Histidine residues 94, 134, 144, 162, 269, 286, 328, 332, 473, 495, 514, 606, and 608 were protonated at the ε position. All other Asp and Glu residues were deprotonated. The LEAP module in Amber 18[54] was used to add the missing hydrogen atoms. The general AMBER force field (GAFF[55]) was used for the substrates, **1** and **3**, while the RESP method[56] was used to calculate the partial atomic charges at HF/6-31 G*. The force field for the iron-containing active site was parameterized using the MCPB[57]. To neutralize the total charge of the system, 46 $Na^+$ ions were added to the surface of the protein, and the resulting protein was solvated with a 15 Å layer of TIP3P[58] waters. We used the Amber ff14SB[59] force field for the protein residues. For system (b) and (c), **3** were docked into the active site using Schrödinger Small-Molecule Drug Discovery Suite 2017-1 (Schrödinger LLC, New York, USA). Supplementary Fig. 36 shows the docking result of system (c).

After the setup, the energy of the system was minimized by 5000 steps of steepest descent and 5000 steps of conjugate gradient methods. Then the system was heated from 0 K to 300 K under a canonical ensemble for 0.5 ns. Subsequently, a 1 ns density equilibration was performed under the NPT ensemble at 300 K and 1.0 atm pressure to get a uniform density after the heating dynamics. Subsequently, we removed all restraints on the protein and further equilibrated the system for 4 ns under an NPT ensemble to obtain the well settled pressure and temperature. Finally, a productive MD run was conducted for 50 ns. During all MD simulations, the covalent bonds containing hydrogen were constrained using SHAKE[60]. All simulations were performed with the GPU version of the Amber 18 package[54].

## QM/MM methodology

A representative structure of MD simulation for system (a) was chosen to do the following QM/MM calculations. All QM/MM calculations were performed using ChemShell[61], combining Turbomole[62] for the QM region and DL_POLY[63] for the MM region. Residues within 12.0 Å of iron center were included into the active region. The electronic embedding[64] scheme was employed to account for the polarizing effect of the enzyme environment on the QM region. Hydrogen link atoms[65] with the charge-shift model[66] were applied to treat the QM/MM boundary. B3LYP[67] has been proved to be a successful functional for studying iron-based metalloenzymes[68], and the oxidation state of +3 has been suggested for Fe cofactor in GdmN based on the study of HypF[15]. During QM/MM calculations, the QM region was studied by B3LYP at two levels of theory. The QM region consists of the truncated **1** and carbamoyl-AMP, and the side chains of His133, His137, Glu156, Glu333 as well as the $Fe^{3+}$ ion as shown in Supplementary Fig. 25b. For geometry optimization, all-electron basis of Def2-SVP (labeled B1) was used. The energies were further corrected with the larger basis set Def2-TZVP (labeled B2). The dispersion correction was included with Grimme's D3BJ[69] method. The DL-FIND[70] optimizer was used in the geometry optimization. The transition states were determined as the highest point on the potential energy surface along the reaction coordinates, for which a small increment of 0.02 Å was used for scanning near the transition states. All transition states (TSs) were located by relaxed potential energy surface (PES) scans followed by the full TS optimizations using the P-RFO optimizer.

## QM calculations

All of the Hybrid cluster-continuum (HCC) model calculations[33,34] were performed with the Gaussian 16 software[71]. The geometries of all of the TSs, reactants, and intermediates involved in the reaction were fully optimized in conjunction with the SMD continuum solvation model[72] at the B3LYP/6-31 G(d) level. Harmonic frequency calculations were performed at the same level of theory as the optimizations in order to estimate the zero-point energies as well as the thermal and entropic corrections. The corrections between the stable structures and the transition states were ascertained by analyzing the corresponding imaginary frequency modes, as well as by limited intrinsic reaction coordinate (IRC) calculations. The energies were further refined at BMK/6-311 + +G(d,p)[73], which has been proven to be reliable for HCC model calculations[74–76].

## Isolation and identification of the compounds

For isolation of 20-*O*-methyl-19-chloroproansamitocin (**1**), a 10-l scale fermentation of WJH10 was carried out. The fermentation broth was centrifuged, and the supernatant was extracted three times with ethyl acetate. Then, the crude extract was subjected to microporous resin XAD-16 (Amberlite), which was washed with 30%, 65%, and 100% ethanol. The fractions were analyzed by HPLC. Fractions containing compound **1** were concentrated and further purified by HPLC on a Thermo BDS HYPERSIL C18 column (250 × 10 mm, 5-μm particle size). For isolation of *N*-demethyl-4,5-desepoxy-maytansinol (**3**), a 10-l scale fermentation of WJH13 was carried out. The fermentation broth was centrifuged, and the supernatant was extracted three times with ethyl acetate. Then, the crude extract was subjected to MPLC over RP-18 silica gel (40–75 mm, 160 g) eluted with $H_2O$, 50%, 75%, and 100% MeOH. The fractions were analyzed by HPLC. Fractions containing compound **3** were concentrated and further purified by HPLC on a Thermo BDS HYPERSIL C18 column (250 × 10 mm, 5-μm particle size). *N*-demethyl-3-carbamoyl-4,5-desepoxy-maytansinol (**4**) was obtained by scale-up of the conversion of **3** catalyzed by the GdmN-V24Y/G157A/R158A/G188R mutant. The large-scale reactions (20 ml) were performed in a reaction mixture containing 10 mM $MgCl_2$, 1 mM carbamoyl phosphate, 2 mM ATP, 0.5 mM **3**, 50 mM Tris, pH 8.5, 200 mM NaCl, and 40 μM proteins at 30 °C for 24 h. After incubation, the reaction mix was diluted by 1:5 in methanol. The sample was further purified by HPLC on a Thermo BDS HYPERSIL C18 column (250 × 10 mm, 5-μm particle size) with a flow rate of 2 ml min⁻¹. The mobile phase consisted of 40% solvent A (water) and 60% solvent B (methanol). The resulting compounds were collected and dried for NMR analysis using MestReNova version 9.0.1 and high-resolution mass spectrometry (HRMS) analysis ($m/z$ 600.2083 [M + Na]⁺ calc., $m/z$ 600.2092 [M + Na]⁺ found).

¹H NMR (400 MHz, CDCl₃): δ 7.93 (s, 1H, H-17), 6.59-6.52 (m, 1H, H-12), 6.55 (s, 1H, H-21), 6.04 (d, J = 10.8, 1H, H-13), 5.57 (dd, J = 15.2, 9.2, 1H, H-11), 5.35 (d, J = 8.8, 1H, H-5), 5.22 (br, s, 1H, H-3), 4.41-4.35 (m, 1H, H-7), 3.90 (s, 3H, H-20-OMe), 3.51 (d, J = 9.2, 1H, H-10), 3.41 (d, J = 14.4, 1H, H-15b), 3.34 (s, 3H, H-10-OMe), 3.17 (d, J = 14.8, 1H, H-15a), 2.88 (br s, 2H, H-1), 2.62-2.57 (m, 1H, H-6), 2.06-1.98 (m, 1H, H-8b), 1.76 (s, 3H, H-14-Me), 1.69 (s, 3H, H-4-Me), 1.32-1.25 (m, 1H, H-8a), 1.15 (s, J = 6.4, 3H, H-6-Me).

¹³C NMR (100 MHz, CDCl₃): δ 156.2 (s, C-3-carbam), 154.7 (s, C-20), 152.9 (s, C-7-carbam), 139.6 (s, C-18), 139.0 (s, C-16), 135.6 (s, C-14), 133.5 (d, C-12), 132.8 (s, C-4), 129.8 (d, C-13), 127.3 (d, C-11), 125.2 (d, C-5), 112.4 (d, C-17), 108.1 (d, C-21), 87.2 (d, C-10), 81.6 (s, C-9), 78.0 (d, C-7), 73.9 (d, C-3), 56.3 (q, C-20-OMe), 56.0 (s, C-10-OMe), 45.5 (t, C-15), 38.2 (t, C-2), 37.7 (d, C-6), 34.3 (t, C-8), 17.8 (q, C-6-Me), 17.3 (q, C-14-Me), 15.2 (q, C-14-Me).

**Crystallization and single-crystal X-ray diffraction analyses of 4**

Crystals of **4** were crystallized from a methanol solution by slow evaporation at room temperature. Data were collected on a Bruker D8 VENTURE CMOS Photon II diffractometer with helios mx multilayer monochromator Cu Kα radiation ($\lambda = 1.54178$ Å) at 173 K. Data collection, unit cell refinement and data reduction were performed using APEX3 v2019.11-0. The structure was solved by Intrinsic Phasing method using SHELXL-2018/3 program package[77], and refined by full-matrix least-squares on F2 with anisotropic displacement parameters for the non-H atoms using SHELXL-2018/3 program package. The crystallographic data have been deposited at the Cambridge Crystallographic Data Centre with accession number 2122658. The detailed statistics are summarized in Supplementary Table 3.

**Size-exclusion chromatography**

The aggregation states of GdmN and mutants in solution were determined by size-exclusion chromatography using a fast protein liquid chromatography system (Bio-Rad) with a Superdex 200 Increase 10/300 GL gel filtration column (Cytiva) pre-equilibrated in buffer C (10 mM Tris, 150 mM NaCl, pH 8.0). The fractions of the highest purity as judged by SDS-PAGE were concentrated to 10 mg ml$^{-1}$. Thyroglobulin (670.0 kDa), gamma globulin (158.0 kDa), ovalbumin (44.0 kDa), myoglobin (17.0 kDa) and vitamin $B_{12}$ (1.3 kDa) from Gel Filtration Standard kits (Bio-Rad) were used for calibration. Blue Dextran was used to determine the column void volume. The calibration curve of $K_{av}$ versus log(MW) was prepared using Eq. (1):

$$K_{av} = (V_e - V_o)/(V_t - V_o) \qquad (1)$$

Where $V_o$ is the column void volume, $V_e$ is the elution volume, and $V_t$ is the total bed volume.

**Reporting summary**

Further information on research design is available in the Nature Research Reporting Summary linked to this article.

## Data availability

Data supporting the findings of this work are available within the paper and its Supplementary Information files. A reporting summary for this Article is available as a Supplementary Information file. X-ray crystallographic coordinates have been deposited in the Protein Data Bank (PDB) with the accession codes 7VYO, 7VX0, 7VYJ, 7VZY, 7VZN, 7VZZ, 7VYP, 7VZU, and 7VZQ. The X-ray crystallographic data for **4** have been deposited at the Cambridge Crystallographic Data Centre with the accession number CCDC 2122658. Copies of the data can be obtained free of charge via https://www.ccdc.cam.ac.uk/structures/. Source data are provided with this paper. Data is available from the corresponding authors upon request. Source data are provided with this paper.

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

## Acknowledgements

The X-ray diffraction data were collected at the Shanghai Synchrotron Radiation Facility, beamline BL18U1 and BL19U1, and we thank the staff at the Shanghai Synchrotron Radiation Facility for assisting X-ray diffraction data collection. We thank the staff of the Core Facility and Technical Service Center for SLSB and the Instrumental Analysis Center in SJTU for data collection. We thank Prof. Jiahai Zhou, Lian Wu, Xin Zang, and Yuxiang Gao for suggestions on protein crystal structure determination, as well as Dr. Ting Shi and Qian Yu for suggestions on computational analysis. We thank Xingzhi Jiao for help in chemical compound structure determination and Dr. Wei Zhang for suggestions on analysis of LC-MS data. This work was supported by The National Key R&D Program of China (2019YFA0905400, 2021YFC2100600), the National Natural Science Foundation of China (31830104), and Science and Technology Commission of Shanghai Municipality (19JC1413000, 19430750600, 17JC1403600).

## Author contributions

L.B. and Q.K. conceived the project. L.B., Q.K., and J.W. designed the experiments. J.W., Y.Z., and X.C. performed the experiments. J.W. and Z.L. analyzed the data. X.Z. and B.W. performed computational studies. J.W., L.B., J.Z., and Q.K. wrote the manuscript. J.W., X.Z., L.B., B. W., and M.T. revised the manuscript.

## Competing interests

The authors declare no competing interests.
