## [Peer Review File · Nature Communications]

Endowing homodimeric carbamoyltransferase GdmN with iterative functions through structural characterization and mechanistic studiesREVIEWER COMMENTS

Reviewer #1 (Remarks to the Author):

This is a thorough and very well written study addressing the rational redesign of GdmN to become an iterative enzyme capable of catalyzing two consecutive carbamoylations on a single substrate, as a strategy to expand the potential and increase the efficiency of synthetic applications. In particular, they aimed for an enzyme capable of performing successive 7-O and 3-O carbamoylations of maytansinoids derivatives. Previously, they had described that Asm21, a carbamoyl transferase within the GdmN group, was capable of a dual carbamoylation during ansamitocin synthesis. The authors undertake an extensive and in-depth structural and functional characterization of these enzymes that will allow them to successfully redesign the active site and broaden the catalytic capabilities.

The strategy employed and the final results are of significance and great interest for rational protein engineering of new catalytic tools.

Starting from the identification of a new subgroup of carbamoyl transferases, the authors develop a careful and complete characterization of the enzyme GdmN. First, they prove that GdmN can catalyze the 7-O carbamoylation of ansamitocin both in vivo, by exogenous expression of GdmN in Asm21-deficient strains of *Actinosynnema pretiosum*, and in vitro, using the purified recombinant enzyme. They show that recombinant GdmN is a homodimer in solution, in contrast to other carbamoyl transferases that are either homotrimers or monomers, and that dimerization and enzymatic activity depend on a specific domain at the C-terminus of the protein. Next, they determine the crystal structure of GdmN revealing a new type of dimer architecture unknown among carbamoyl transferases. The high-quality crystal structures, 9 in total, including apo form and complexes with substrates and reaction intermediates, allow the authors to provide very detailed descriptions of substrate-binding and reactions mechanisms. The structural claims are well-supported by extensive site-directed mutagenesis in combination with activity assays. The results show significant differences with TobZ the other carbamoyltransferase from this subgroup for which the structure was available. Next, and guided by the new structural knowledge, the authors model the structure of Asc21b, responsible for the 3-O carbamoylation reaction that the authors aim to obtain as iterative reaction in GdmN. Based on the predicted structural differences, the author performed single and multiple mutations in GdmN to redirect the active site to perform successive 7-O and 3-O carbamoylations of the substrate. The formation of the desired compound is confirmed by NMR, mass spectrometry and X-ray crystallography. They even determine the crystal structure of the re-designed iterative GdmN enzyme free and bound to one of the intermediate substrates, verifying the increased plasticity of the reprogrammed active site. Finally, sequence analysis and homology modeling reveal residues in this subgroup of carbamoyltransferases that could account for they different specificity and that could be altered to increase substrate flexibility. The authors also perform computational analysis for the cyclization of the product and for its binding to the mutated enzyme, which shows that they do not leave any loose ends in the story, but I cannot comment on this computational analysis because is not my expertise.

Overall, the article compiles an impressive amount of work that combines many different and sound methodologies. The results are solid, clearly described and exhaustively documented (very extensive supplementary material). The conclusions of the manuscript are well supported by the data and analysis of the authors.

I strongly support the acceptance and publication of the article, and only have some minor questions/suggestions:

- page 6, line 107: In this sentence, for clarity, the authors could specify that they refer to Class III CTases: "The holoenzymes of all characterized (class III) CTases..."

- page 9, line 157: Authors might want to consider replacing "FPLC analysis" by "size-exclusion chromatography analysis".

- It might be illustrative to show a superposition (with the rmsd calculation) of the 2 subunits of GdmN in the asymmetric unit.

- page 12, line 212: It is said that the enzyme contains an iron ion. Perhaps the authors could briefly explain the evidence of the presence of this ion. Distances and geometry of coordination sphere? Similarity with TobZ or other enzymes? Perhaps figure S14 could be completed with the electron density map in the iron coordination sphere.

- Figure 3. Perhaps indicating the position of the iron ions with a sphere of different color will help to better identify the location of the two active sites in the dimer.

- page 15, line 260-261. "The movements of the side chains of V24 and F25 resulted in better shape complementarity to accommodate 1". In figure 4, these movements appear very small, perhaps less than 1 Å. Perhaps the authors might want to specify the distance between the position of the C-alphas of the residues in the two structures to give a better sense of this movement.

- page 19, line 349. Perhaps put "asparagine" or "Asn" instead of "N" to avoid confusion?

- I am curious about the role of Mg²⁺ in the enzyme reaction. The concentration of MgCl₂ for crystallization is 1.5 mM, which seems small for the combination with 1.5 mM ATP and 1.5 mM CP. In contrast, for the activity assays, the MgCl₂ concentration is 10 mM. Why is Mg²⁺ important? Why the structures do not show the bound ion?

- Figure S19, legend. It is not clear what the authors mean with "Stereo" 2Fo-Fc maps....

- The authors might want to consider changing the order of the figures in the supplementary material to follow the order in which they appear in the text. For instance, figure S10 appears before than figures S6–9; S13 is mentioned earlier than S11 and S12; S21 appears before than S20; and S35 and S36 appear before than S29–34. I also suggest to merge figures S11 and S13, perhaps eliminating panel S13a, since it already shows in Fig. 3. Also, table S4 appears in the text before than tables S1–3.

Reviewer #2 (Remarks to the Author):

In this work, a carbamoyltransferase GdmN was engineered based on several holo-crystal structures to gain iterative function of successive 7-O and 3-O carbamoylation, in which hydrogen bonding interaction and loop flexibility were found to be critical to the regioselectivity. The manuscript is overall well organized and supported by solid data. There are several points to be clarified.

1. Based on the crystal structures obtained, MD simulation would be an effective approach for elucidating the catalytic activity and mechanisms. Did authors performed any MD? for example, compound 3 and GdmN tetra mutant (Fig S33).

2. How about the effect of key residues identified in this study on other CTases? it also would be interesting to interrogate the corresponding residues in Asm21.

3. What is the reaction time of Fig 6? will 3 be completely converted to 4 in the end?

4. Line458 Y82 and F82?

Reviewer #3 (Remarks to the Author):

This manuscript describes some interesting enzyme "redesign" but there are major issues that must be addressed before publication.

1) Why are the authors convinced that they have an X-ray structure of a tetrahedral intermediate? The 2Fo-Fc omit maps in Figure S19 are not at all convincing.

2) How do the authors know that tetrahedral intermediate decomposition is rate-limiting? I see no convincing evidence of that.

3) The process for making this catalyst *iterative* does not seem rational to me. Why should readers believe it was not an accidental discovery? What are the principles that make a catalyst "iterative" and how were those used in the *design* of mutants?

4) No references are provided for DFT computational methods. Also, the barriers reported are all too high for biological conditions for the ring closing nucleophilic attack examined.

Response to Reviewers' Comments

We sincerely thank the reviewers for the insightful comments and constructive suggestions to improve the quality of our manuscript. We do agree with all comments from reviewers and have carefully addressed all the reviewers' comments. We provide a point-by-point response to all comments from reviewers in **BLUE** font below (the line numbers and figure numbers refer to the revised manuscript unless otherwise stated). Moreover, the revised contents are in **RED**, and AMP-CP, the previous abbreviation for carbamoyladenylate, was accurately changed to carbamoyl-AMP throughout the manuscript.

Reviewer #1 (Remarks to the Author):

This is a thorough and very well written study addressing the rational redesign of GdmN to become an iterative enzyme capable of catalyzing two consecutive carbamoylations on a single substrate, as a strategy to expand the potential and increase the efficiency of synthetic applications. In particular, they aimed for an enzyme capable of performing successive 7-O and 3-O carbamoylations of maytansinoids derivatives. Previously, they had described that Asm21, a carbamoyl transferase within the GdmN group, was capable of a dual carbamoylation during ansamitocin synthesis. The authors undertake an extensive and in-depth structural and functional characterization of these enzymes that will allow them to successfully redesign the active site and broaden the catalytic capabilities.

The strategy employed and the final results are of significance and great interest for rational protein engineering of new catalytic tools.

Starting from the identification of a new subgroup of carbamoyl transferases, the authors develop a careful and complete characterization of the enzyme GdmN. First, they prove that GdmN can catalyze the 7-O carbamoylation of ansamitocin both in vivo, by exogenous expression of GdmN in Asm21-deficient strains of *Actinosynnema*

pretiosum, and in vitro, using the purified recombinant enzyme. They show that recombinant GdmN is a homodimer in solution, in contrast to other carbamoyl transferases that are either homotrimers or monomers, and that dimerization and enzymatic activity depend on a specific domain at the C-terminus of the protein. Next, they determine the crystal structure of GdmN revealing a new type of dimer architecture unknown among carbamoyl transferases. The high-quality crystal structures, 9 in total, including apo form and complexes with substrates and reaction intermediates, allow the authors to provide very detailed descriptions of substrate-binding and reactions mechanisms. The structural claims are well-supported by extensive site-directed mutagenesis in combination with activity assays. The results show significant differences with TobZ, the other carbamoyltransferase from this subgroup for which the structure was available. Next, and guided by the new structural knowledge, the authors model the structure of Asc21b, responsible for the 3-*O* carbamoylation reaction that the authors aim to obtain as iterative reaction in GdmN. Based on the predicted structural differences, the author performed single and multiple mutations in GdmN to redirect the active site to perform successive 7-*O* and 3-*O* carbamoylations of the substrate. The formation of the desired compound is confirmed by NMR, mass spectrometry and X-ray crystallography. They even determine the crystal structure of the re-designed iterative GdmN enzyme free and bound to one of the intermediate substrates, verifying the increased plasticity of the reprogrammed active site. Finally, sequence analysis and homology modeling reveal residues in this subgroup of carbamoyltransferases that could account for their different specificity and that could be altered to increase substrate flexibility. The authors also perform computational analysis for the cyclization of the product and for its binding to the mutated enzyme, which shows that they do not leave any loose ends in the story, but I cannot comment on this computational analysis because it is not my expertise.

Overall, the article compiles an impressive amount of work that combines many different and sound methodologies. The results are solid, clearly described and exhaustively documented (very extensive supplementary material). The conclusions of the manuscript are well supported by the data and analysis of the authors.

I strongly support the acceptance and publication of the article, and only have some minor questions/suggestions:

- page 6, line 107: In this sentence, for clarity, the authors could specify that they refer to Class III CTases: “The holoenzymes of all characterized (class III) CTases...”

Response: We have changed the sentence to “The holoenzymes of all characterized (class III) CTases...” as suggested.

- page 9, line 157: Authors might want to consider replacing “FPLC analysis” by “size-exclusion chromatography analysis”.

Response: We have changed all “FPLC” to “size-exclusion chromatography” as suggested.

- It might be illustrative to show a superposition (with the rmsd calculation) of the 2 subunits of GdmN in the asymmetric unit.

Response: We have added “The monomer is similar to the other one, with an r.m.s.d. value of ~ 0.16 Å” in line 186 of page 10. Besides, we also showed the superposition of the 2 subunits of GdmN in the asymmetric unit in **Supplementary Figure S8b** in the revised supplementary material as suggested.

- page 12, line 212: It is said that the enzyme contains an iron ion. Perhaps the authors could briefly explain the evidence of the presence of this ion. Distances and geometry of coordination sphere? Similarity with TobZ or other enzymes? Perhaps figure S14 could be completed with the electron density map in the iron coordination sphere.

Response: Similar with TobZ complexed with carbamoyl-AMP (PDB: 3VER) and HypF complexed with the carbamoyl-AMP analog AMPCP (PDB: 3VTH), iron ion in GdmN showed standard octahedral coordination caused by interactions with the α -phosphate group of the carbamoyl-AMP molecule, the amine of the carbamoyl group from the carbamoyl-AMP molecule, two histidines, and two aspartates (**Figure r1**). Besides, we also added *2Fo-Fc* map for iron ion and its coordinating residues contoured

at 1.0 σ in **Supplementary Figure S10** in the revised supplementary material as suggested.

Figure r1 The conserved structural features coordinating iron ion in the structure of GdmN in complex with carbamoyl-AMP (a), TobZ in complex with carbamoyl-AMP (b), and HypF in complex with the carbamoyl-AMP analog AMPCP (c).

- Figure 3. Perhaps indicating the position of the iron ions with a sphere of different color will help to better identify the location of the two active sites in the dimer.

Response: We have adjusted **Figure 3** in the revised manuscript as suggested.

- page 15, line 260-261. “The movements of the side chains of V24 and F25 resulted in better shape complementarity to accommodate **1**”. In figure 4, these movements appear very small, perhaps less than 1 Å. Perhaps the authors might want to specify the distance between the position of the C-alphas of the residues in the two structures to give a better sense of this movement.

Response: We have specified the distance between the position of the side chains of the residues in the two structures in **Figure 4** in the revised manuscript.

- page 19, line 349. Perhaps put “asparagine” or “Asn” instead of “N” to avoid confusion?

Response: We have replaced one-letter codes with three-letter codes of amino acids in the revised manuscript as suggested.

- I am curious about the role of Mg^{2+} in the enzyme reaction. The concentration of $MgCl_2$ for crystallization is 1.5 mM, which seems small for the combination with 1.5 mM ATP and 1.5 mM CP. In contrast, for the activity assays, the $MgCl_2$ concentration is 10 mM. Why is Mg^{2+} important? Why the structures do not show the bound ion?

Response: The CTases occurred in natural products catalyze two nucleophilic half-reactions: the YrdC-like domain catalyzes the formation of carbamoyl-adenylate (carbamoyl-AMP) in the presence of CP, ATP, and Mg^{2+} ; the Kae1-like domain transfers the carbamoyl group from carbamoyl-AMP to natural products¹. Mg^{2+} ion plays an important role in the carbamoyl-AMP formation. *In vitro* studies on Asm21 and NovN confirmed intriguing Mg^{2+} dependency^{2,3}. To interpret the role of Mg^{2+} ion, we have compared the structure of the YrdC-like domain of GdmN with that of TobZ. The structure of the YrdC-like domain is similar to that of TobZ, with an r.m.s.d. value of 0.73 Å for 163 Cα atoms (**Supplementary Figure S17**). The structural superposition revealed that the site coordinating the Mg^{2+} ion is highly conserved, with conservation of residues corresponding to S533 in GdmN and S530 in TobZ. Further substituting S533 with Ala abolished the catalytic activity. The above experimental results determined that Mg^{2+} ion is important.

In the previous study², we have examined different concentrations of different metal ions on the catalytic activity, and found that the best condition is 10 mM $MgCl_2$. However, when we carried out crystallization experiments, we found that 10 mM $MgCl_2$ easily generates magnesium salt crystals in the crystallization buffer, affecting the generation of protein crystals. Through several trials, we found that protein crystals easily appear in the crystallization buffer containing 1.5 mM $MgCl_2$.

Although Mg^{2+} ion was included in the crystallization buffer, no electron density

was observed at the position of Mg²⁺ in the structures. Similar conditions appeared in other crystallographic studies of other Mg²⁺-dependent enzymes^{4,5}. In crystallographic study of the ATP-binding cassette (ABC) superfamily of ATPase (UvrA)⁴, although MgCl₂ was added in the crystallization buffer, there was no electron density of Mg²⁺ at the position adjacent to the β-phosphate of ADP, which was normally occupied in the structures of ABC ATPases complexed with Mg²⁺-ADP. In crystallographic study of *Bacillus anthracis* dihydropteroate synthase (BaDHPS)⁵, the crystals were soaked in the crystallization buffer containing MgCl₂, while no electron density of Mg²⁺ was observed in the structures.

- Figure S19, legend. It is not clear what the authors mean with “Stereo” 2Fo-Fc maps....

Response: We are sorry for this. We referred to Figures S4-S6 in the literature¹, and intended to mean the stereo view of 2Fo-Fc maps. Thanks to reviewer’s reminder, we referred to other literatures^{6,7}, and found that this word is not suitable. In order to avoid confusion, we have removed “Stereo” in the revised supplementary material.

- The authors might want to consider changing the order of the figures in the supplementary material to follow the order in which they appear in the text. For instance, figure S10 appears before than figures S6–9; S13 is mentioned earlier than S11 and S12; S21 appears before than S20; and S35 and S36 appear before than S29–34. I also suggest to merge figures S11 and S13, perhaps eliminating panel S13a, since it already shows in Fig. 3.

Also, table S4 appears in the text before than tables S1–3.

Response: We have changed the order of the figures and tables in the revised supplementary material as suggested.

Reviewer #2 (Remarks to the Author):

In this work, a carbamoyltransferase GdmN was engineered based on several holo-crystal structures to gain iterative function of successive 7-*O* and 3-*O* carbamoylation, in which hydrogen bonding interaction and loop flexibility were found to be critical to the regioselectivity. The manuscript is overall well organized and supported by solid data. There are several points to be clarified.

1. Based on the crystal structures obtained, MD simulation would be an effective approach for elucidating the catalytic activity and mechanisms. Did authors perform any MD? for example, compound **3** and GdmN tetra mutant (Fig S33).

Response: We appreciate the instructive suggestions. We agree with the reviewer that MD simulation could help us to further understand the occurrence of 3-*O* carbamoylation of compound **3** catalyzed by GdmN tetra mutant. Following the suggestions, we have performed MD simulations for both the wild-type GdmN and the tetra-mutant GdmN, where compound **3** was docked into the active sites. For the wild-type GdmN in complex with compound **3**, our MD simulations showed that compound **3** maintains a stable but unfavorable conformation for 3-*O* carbamoylation, in which the O3 of compound **3** is far away from the C1 of carbamoyl-AMP, with an average O3—C1 distance of $\sim 8\text{\AA}$ (**Supplementary Figure S37**). In contrast to the wild-type GdmN, our MD simulations of the tetra-mutant GdmN showed that O3 of compound **3** is in close proximity to the C1 of carbamoyl-AMP, with an average O3—C1 distance of $\sim 4\text{\AA}$ (**Supplementary Figure S38**). Such distance is favorable for the 3-*O* carbamoylation reaction. The relevant discussions have been added into the revised manuscript.

2. How about the effect of key residues identified in this study on other CTases? it also would be interesting to interrogate the corresponding residues in Asm21.

Response: Thanks for the suggestions.

We have carried out sequence alignment and homology modeling of other class III CTases (**Supplementary Figure S4 and S39**). We found that the active site histidine

(His27 in GdmN corresponds to His27 in Asm21) is highly conserved. The residues responsible for the formation of carbamoyl-adenylate (carbamoyl-AMP) are highly conserved, with conservation of residues corresponding to K443/M476/S533 in GdmN and K446/M479/S536 in Asm21. The “keystone” tyrosine, which stabilizes the substrate binding pocket (Y82 in GdmN corresponds to Y82 in Asm21) is highly conserved across class III CTases. On the basis of biochemical experiments and crystallographic studies in our study, we preliminarily speculated that these residues play similar roles in other CTases.

As shown in **Supplementary Figure S1**, these CTases recognize different substrates. Further sequence alignment of substrate binding pocket revealed subtle alterations in “plasticity residues” (**Supplementary Figure S39**), which may contribute to the considerable substrate flexibility of these CTases. It will be intriguing to decipher whether multi-site mutagenesis of these residues in Asm21, Asc21b, and other CTases, could repurpose substrate flexibility. We will undertake further structure-guided engineering and crystallographic studies of Asm21, Asc21b, and other CTases, thereby better interpreting the effect of these residues on other CTases.

3. What is the reaction time of Fig 6? will **3** be completely converted to **4** in the end?

Response: The activity assays of GdmN tetra mutant were performed at 30 °C for 24 h, to yield **3:4** in roughly a 2:1 ratio at 39.85% conversion, as mentioned in the legend of **Figure 6**. We have extended the reaction time of this reaction system, while compound **3** could not be completely converted to compound **4**. When we reduced the concentration of compound **1** to 0.1 mM and kept other conditions unchanged, compound **1** could be completely converted to compound **4** (**Figure r2**). In order to keep the reaction system of the study consistent and show the successive reaction process, we have put the results in **Figure 6**.

Figure r2 HPLC results showing conversion of 1 to 4 by the mutant. The activity assays of the mutant were performed with 1, CP, ATP, and MgCl₂ at 30 °C for 24 h.

4. Line 458 Y82 and F82?

Response: Sorry for this, we have corrected it in the revised manuscript.

Reviewer #3 (Remarks to the Author):

This manuscript describes some interesting enzyme "redesign" but there are major issues that must be addressed before publication.

1) Why are the authors convinced that they have an X-ray structure of a tetrahedral intermediate? The $2Fo-Fc$ omit maps in Figure S19 are not at all convincing.

Response: Thanks for the suggestions. The $Fo-Fc$ omit maps of different ligands have been added in **Supplementary Figures S15, S18, and S34**. We also re-drawn the $2Fo-Fc$ maps of different ligands for clarity in **Supplementary Figures S16, S19, and S34**. Both $Fo-Fc$ omit maps and $2Fo-Fc$ maps showed the continuous electron density in the KaeI-like domain, implying the formation of the tetrahedral intermediate. Besides, we have performed QM/MM calculations to further support the formation of the tetrahedral intermediate in **Supplementary Figure S25**.

2) How do the authors know that tetrahedral intermediate decomposition is rate-limiting? I see no convincing evidence of that.

Response: The comment is insightful and helpful. Previously, we assumed that the fracture of the tetrahedral intermediate might be rate-limiting based on the crystal structure in complex with the tetrahedral intermediate. However, we had no evidence to support these assumptions. Thanks to reviewer's reminder, we have carried out QM/MM calculations on enzymatic reactions catalyzed by GdmN to get more insights into the catalytic mechanism of GdmN.

In previous studies of HypF⁸, the oxidation state of +3 has been suggested for Fe cofactor. In this study, we have utilized computational analysis to compare the reactivity for Fe(III) and Fe(II) in the enzymatic reactions. For Fe(III) state, our QM/MM calculations showed that the enzymatic reaction from compound **1** to compound **2** requires a barrier of 17.7 kcal/mol (**Supplementary Figure S25**). However, for Fe(II), the QM/MM calculations (**Figure r3**) show that the tetrahedral intermediate (Int1') is already 29.4 kcal/mol higher than RC' in energy, suggesting that Fe(II) may be inert for the reaction. Such prediction is in line with the spectroscopy study of HypF⁸, showing

that the high oxidation state of Fe(III) is required for the catalysis of GdmN. For the conversion from compound **2** to compound **3** in water, we re-evaluated the reaction barrier with the more reliable cluster-continuum model calculations (**Supplementary Figures S22 and S23**). Our calculations showed that this conversion of compound **2** in water is relatively facile, with a Gibbs energy barrier of 15.3 kcal/mol. Thus, all these calculations tend to support that the decomposition of tetrahedral intermediate could not be the rate-limiting step. Instead, the rate-limiting step could be the enzymatic reaction for the first step of 7-*O* carbamoylation reaction (**Supplementary Figure S25**). According to these results, we were lucky enough to gain the crystal structures complexed with the intermediate. Based on these results, we have revised the manuscript, and detailed calculation results have been added to the revised supplementary material.

Figure r3 (a) QM(UB3LYP/B2)/MM relative energies (kcal/mol) for the enzymatic reaction from **1** to **2** for the Fe(II). The dispersion corrections are included in the relative energies; (b) QM(UB3LYP/B1)/MM-optimized geometries of key species involved in the reaction. Key distances are given in Å. The green colored structure is the truncated structure of the substrate **1**. For clarity, the protein scaffold was omitted.

3) The process for making this catalyst *iterative* does not seem rational to me. Why should readers believe it was not an accidental discovery? What are the principles that make a catalyst "iterative" and how were those used in the *design* of mutants?

Response: We understand the concern from the reviewer. In recent years, some iterative enzymes were reported, such as methyltransferases⁹, glycosyltransferases¹⁰⁻¹², and etc. However, the transition processes from canonical enzymes to iterative enzymes remain poorly understood. To explore the processes, some researchers attempted to utilize protein engineering to repurpose canonical enzymes into iterative enzymes in recent years¹³. Xiaojing Wang and coworkers evaluated two orthologous *O*-methyltransferases (58% identity), which modify the same substrate to afford orthogonal regioisomeric outcomes. Based on protein homology modeling, they proposed that subtle alternations of residues lining the substrate binding pocket force the substrates to adopt different binding poses. Subsequently, structure-guided engineering of these “plasticity residues” showed that site-specific replacement of these residues successfully reprogrammed the regioselectivity of the enzyme to create iterative mutants.

Inspired by these studies, we examined ansamycins and their corresponding CTases (**Supplementary Figure S1**). During ansamitocin biosynthesis, Asm21 could catalyze dual carbamoylation, utilizing both a polyketide backbone and a glycosyl moiety as substrates, suggesting that this subclade of CTases might contain a spacious binding pocket and hold promising potential for redesign. In the biosynthesis of ansacarbamitocins, there are two carbamoyltransferase genes, *asc21a* and *asc21b*, responsible for the three carbamoylation modifications, and Asc21b was proved to catalyze 3-*O*-carbamoylation of **3**¹⁴. In our study, we found that GdmN could catalyze **1** to generate **3**. Similar to two orthologous *O*-methyltransferases in the abovementioned studies, GdmN and Asc21b share high sequence identity (61%). Therefore, we performed structural analysis and structure-guided mutagenesis to reprogram the regioselectivity of GdmN. Coupled with homology modeling of Asc21b, we examined differences of substrate-binding pocket of GdmN and Asc21b, and further

performed structure-guided mutagenesis, thereby expanding the substrate scope of GdmN.

Although these successful studies might provide some insights into the transition processes between canonical enzymes and iterative ones, the principles for design of iterative catalysts still required more examples. We will undertake further researches to explore whether structural analysis of enzymes with orthologous regioselectivity and subsequent structure-guided active site engineering can be successfully applied in protein engineering of other iterative enzymes.

4) No references are provided for DFT computational methods. Also, the barriers reported are all too high for biological conditions for the ring closing nucleophilic attack examined.

Response: Thanks for the suggestions. For the conversion from compound **2** to compound **3** in water, we re-evaluated the reaction barrier with the more reliable cluster-continuum model calculations where the central solute is solvated explicitly by 8 water molecules, and the resulting cluster was treated by a dielectric continuum model (**Supplementary Figures S22 and S23**). Our calculations showed that this conversion of compound **2** in water is relatively facile, with a Gibbs free energy barrier of 15.3 kcal/mol. Based on these results, we have revised the manuscript, and detailed calculation results have been added to the revised supplementary material. Besides, we have cited references 33,34, and 71-76 in the revised manuscript.

Reference

1. Parthier, C. *et al.* The *O*-carbamoyltransferase TobZ catalyzes an ancient enzymatic reaction. *Angew. Chem. Int. Ed. Engl.* **51**, 4046-4052 (2012).
2. Li, Y. *et al.* Dual carbamoylations on the polyketide and glycosyl moiety by Asm21 result in extended ansamitocin biosynthesis. *Chem. Biol.* **18**, 1571-1580 (2011).
3. Freel Meyers, C.L. *et al.* Characterization of NovP and NovN: completion of novobiocin biosynthesis by sequential tailoring of the noviosyl ring. *Angew. Chem. Int. Ed. Engl.* **43**, 67-70 (2004).
4. Pakotiprapha, D. *et al.* Crystal structure of *Bacillus stearothermophilus* UvrA provides insight into ATP-modulated dimerization, UvrB interaction, and DNA binding. *Mol. Cell* **29**, 122-133 (2008).

-
5. Yun, M.K. *et al.* Catalysis and sulfa drug resistance in dihydropteroate synthase. *Science* **335**, 1110-1114 (2012).
 6. Wright, N.J. & Lee, S.Y. Structures of human ENT1 in complex with adenosine reuptake inhibitors. *Nat. Struct. Mol. Biol.* **26**, 599-606 (2019).
 7. Husain, N. *et al.* Structural basis for the methylation of G1405 in 16S rRNA by aminoglycoside resistance methyltransferase Sgm from an antibiotic producer: a diversity of active sites in m⁷G methyltransferases. *Nucleic Acids Res.* **38**, 4120-4132 (2010).
 8. Shomura, Y. & Higuchi, Y. Structural basis for the reaction mechanism of S-carbamoylation of HypE by HypF in the maturation of [NiFe]-hydrogenases. *J. Biol. Chem.* **287**, 28409-28419 (2012).
 9. Bat-Erdene, U. *et al.* Iterative catalysis in the biosynthesis of mitochondrial complex II inhibitors harzianopyridone and atpenin B. *J. Am. Chem. Soc.* **142**, 8550-8554 (2020).
 10. Ito, T., Fujimoto, S., Suito, F., Shimosaka, M. & Taguchi, G. C-glycosyltransferases catalyzing the formation of di-C-glucosyl flavonoids in citrus plants. *Plant J.* **91**, 187-198 (2017).
 11. Chen, D. *et al.* Probing and engineering key residues for bis-C-glycosylation and promiscuity of a C-glycosyltransferase. *ACS Catal.* **8**, 4917-4927 (2018).
 12. Zhang, M. *et al.* Functional characterization and structural basis of an efficient di-C-glycosyltransferase from *Glycyrrhiza glabra*. *J. Am. Chem. Soc.* **142**, 3506-3512 (2020).
 13. Wang, X. *et al.* Rational reprogramming of O-methylation regioselectivity for combinatorial biosynthetic tailoring of benzenediol lactone scaffolds. *J. Am. Chem. Soc.* **141**, 4355-4364 (2019).
 14. Li, X., Wu, X. & Shen, Y. Identification of the bacterial maytansinoid gene cluster *asc* provides insights into the post-PKS modifications of ansacarbamitocin biosynthesis. *Org. Lett.* **21**, 5823-5826 (2019).

REVIEWERS' COMMENTS

Reviewer #1 (Remarks to the Author):

The authors have addressed correctly all my comments and I have no further concerns and recommend the publication of the manuscript.

Reviewer #2 (Remarks to the Author):

The authors have revised the manuscript following the reviewer's comments and addressed the concerns raised. I agree the acceptance of this work.

Reviewer #3 (Remarks to the Author):

While I am not completely convinced by the results, I appreciate the authors' hard work in addressing my concerns and those of the other referees. I believe the work is suitable now for publication where readers can form their own opinions.

Manuscript ID: NCOMMS-22-06189-T

Response to Reviewers' Comments

We sincerely thank the reviewers for the insightful comments and constructive suggestions to improve the quality of our manuscript. We do agree with all comments from reviewers and have carefully addressed all the reviewers' comments. Moreover, the revised contents in the manuscript are in **RED**.

Reviewer #1 (Remarks to the Author):

The authors have addressed correctly all my comments and I have no further concerns and recommend the publication of the manuscript.

Response: We sincerely appreciate reviewer's comments and help on improving the quality of the manuscript.

Reviewer #2 (Remarks to the Author):

The authors have revised the manuscript following the reviewer's comments and addressed the concerns raised. I agree the acceptance of this work.

Response: We appreciate the instructive suggestions, esp. on the computational calculation, which helped us to gain better understandings on the catalytic mechanism.

Reviewer #3 (Remarks to the Author):

While I am not completely convinced by the results, I appreciate the authors' hard work in addressing my concerns and those of the other referees. I believe the work is suitable now for publication where readers can form their own opinions.

Response: Thanks for the appreciation of our efforts on structure biology and computational analysis and those insightful and helpful suggestions, which made our data clearer and hypothesis more convincing.